# Realizing tight-binding Hamiltonians using site-controlled coupled cavity arrays

Abhi Saxena [1] ✉, Arnab Manna[2], Rahul Trivedi[1] & Arka Majumdar [1,2] ✉

Analog quantum simulators rely on programmable and scalable quantum devices to emulate Hamiltonians describing various physical phenomenon. Photonic coupled cavity arrays are a promising alternative platform for realizing such simulators, due to their potential for scalability, small size, and high-temperature operability. However, programmability and nonlinearity in photonic cavities remain outstanding challenges. Here, using a silicon photonic coupled cavity array made up of 8 high quality factor ($Q$ up to $\sim 7.1 \times 10^4$) resonators and equipped with specially designed thermo-optic island heaters for independent control of cavities, we demonstrate a programmable photonic cavity array in the telecom regime, implementing tight-binding Hamiltonians with access to the full eigenenergy spectrum. We report a $\sim 50\%$ reduction in the thermal crosstalk between neighboring sites of the cavity array compared to traditional heaters, and then present a control scheme to program the cavity array to a given tight-binding Hamiltonian. The ability to independently program high-Q photonic cavities, along with the compatibility of silicon photonics to high volume manufacturing opens new opportunities for scalable quantum simulation using telecom regime infrared photons.

Achieving analog quantum simulation necessitates the realization of programmable quantum devices[1]. Due to their inherent driven-dissipative nature, photonic systems are a promising platform for non-equilibrium quantum simulation[2]. An archetypal photonic quantum simulator consists of an array of programmable non-linear nodes with access to the entire quantized eigenenergy spectra of the Hamiltonians being simulated. While there have been numerous works on analog quantum simulation with microwave photons[3–7], optical photons with their extremely weak interaction with the environment, can provide several additional advantages. The higher energy of optical photons allows for the preservation of quantum states even at room temperature, enabling operability at much higher temperatures[8], which significantly simplifies the experiments and lowers the resources needed to scale the simulator. Additionally, availability of single photon detectors in the optical domain allows direct measurement of multiparticle correlations[9,10] which are a key set of measurements for characterizing the quantum-ness of realized states. Building on recent advancements in nanofabrication, quantum

optical systems have shifted from bulky tabletop systems prone to misalignments to fully integrated on chip photonic circuits. These large scale photonic integrated circuits owing to their small size and high speed of operation present opportunities for unprecedented scalability to practical quantum advantage[11].

One solution to engineer such quantum systems in optics is via photonic coupled cavity arrays (CCA)[12] where coupling between cavities provides a potential map for photons to move around, and strong spatial confinement of light for long durations allows access to onsite non-linearity via coupling with various excitonic materials. For photonic CCAs to be used as quantum simulators, four broad requirements need to be satisfied, namely, (i) scalability: there must exist pathways to scale to a large number of sites; (ii) measurability: there is a need for protocols to perform Hamiltonian tomography with restricted access and have CCAs with addressability to all the eigenstates of the system; (iii) controllability: control over all the terms describing the Hamiltonian is required; and finally (iv) optical nonlinearity: need to realize photon-photon interaction to simulate many-

[1]Department of Electrical & Computer Engineering, University of Washington, Seattle, WA 98195, USA. [2]Department of Physics, University of Washington, Seattle, WA 98195, USA. ✉e-mail: abhi15@uw.edu; arka@uw.edu

body Hamiltonians. The last demand as a precondition, necessitates using high-quality factor (Q) cavities with small mode volumes as constituents of the CCA. Such high-Q cavities are also necessary to probe the entire quantized eigenenergy spectra. Though several experiments showing various physical phenomena using optical CCAs have been previously reported[13–15], none of these CCAs are programmable and have access to the entire quantized eigenenergy spectra of the Hamiltonian. While careful selection of the operation regime can lead to pathways that allow scalability to multiple sites using photonics[16], in the optical regime achieving programmability and measurability of the eigen-spectrum, is very challenging owing to the extremely small physical dimensions involved.

In this work, we tackle these problems by engineering a silicon photonic CCA made of high-Q (intrinsic $Q$s up to $\sim 7.1 \times 10^4$) racetrack resonators with thermally controllable onsite potential using specially designed thermo-optic (TO) island heaters. Here, we specifically focus on 1D tight-binding lattices which can be described by a set of Gaussian Hamiltonians of the form ($\hbar = 1$):

$$H = \sum_n \mu_n a_n^\dagger a_n + J_n(a_{n+1}^\dagger a_n + a_n^\dagger a_{n+1}) \qquad (1)$$

where $a_n$ denotes the onsite photonic annihilation operator, $\mu_n$ is the onsite potential given by the resonant frequency of the cavity, $J_n$ is the photonic hopping rate between $n^{th}$ and $(n+1)^{th}$ sites. Realization of such a set of Hamiltonians requires implementing a potential profile $[\mu_n] = [\mu_0, \mu_1, \ldots \mu_{N-1}]$ across a photonic lattice with specific inter-site hopping rates $J_n$, while ensuring that all the eigenstates of the system denoted by $[\epsilon_n] = [\epsilon_0, \epsilon_1, \ldots \epsilon_{N-1}]$ remain addressable and measurable.

## Results

### Design and characterization of the optical layer

Experimentally, we implement a Hamiltonian with 8 nodes via a CCA made up of 8 strongly coupled racetrack resonators fabricated on a silicon-on-insulator platform using $220 nm$ silicon on top of $3\,\mu m$ thick silicon oxide (Fig. 1a). The spacing between the resonators is determined by the desired hopping rate between the sites for the tight-binding Hamiltonians being implemented (see Supplementary Information: Section S1 for optical mode profiles and hopping rate calculation). The spectrum of the resulting system is probed via a set of grating couplers located at the first and last sites. The scattering properties of this system are completely described by the effective non-Hermitian Hamiltonian which incorporates the coupling to input/output ports and system losses as:

$$H_{eff}^0 = H - j\left(\frac{\gamma_0}{2} a_0^\dagger a_0 + \frac{\gamma_{N-1}}{2} a_{N-1}^\dagger a_{N-1}\right) - j \sum_n \frac{\kappa_n}{2} a_n^\dagger a_n \qquad (2)$$

where $\gamma_0, \gamma_{N-1}$ denote the coupling rates to the grating couplers and $\kappa_n$ denotes the onsite scattering/absorption losses. Photonic CCAs on a 2D chip can only be accessed reliably through input/output ports at the device boundaries and hence require a protocol that allows determination of the realized Hamiltonians with this boundary-restricted access. To map the initial Hamiltonian $H_{eff}^0$ of our CCA post-fabrication, we extend the Hamiltonian tomography algorithm developed for 1D lossless lattices[17,18] for application in 1D nearest neighbor lossy CCAs (see Supplementary Information: Section S2 for details on the tomography algorithm). The modified algorithm allows for determining the entire $H_{eff}^0$ describing the system from a single

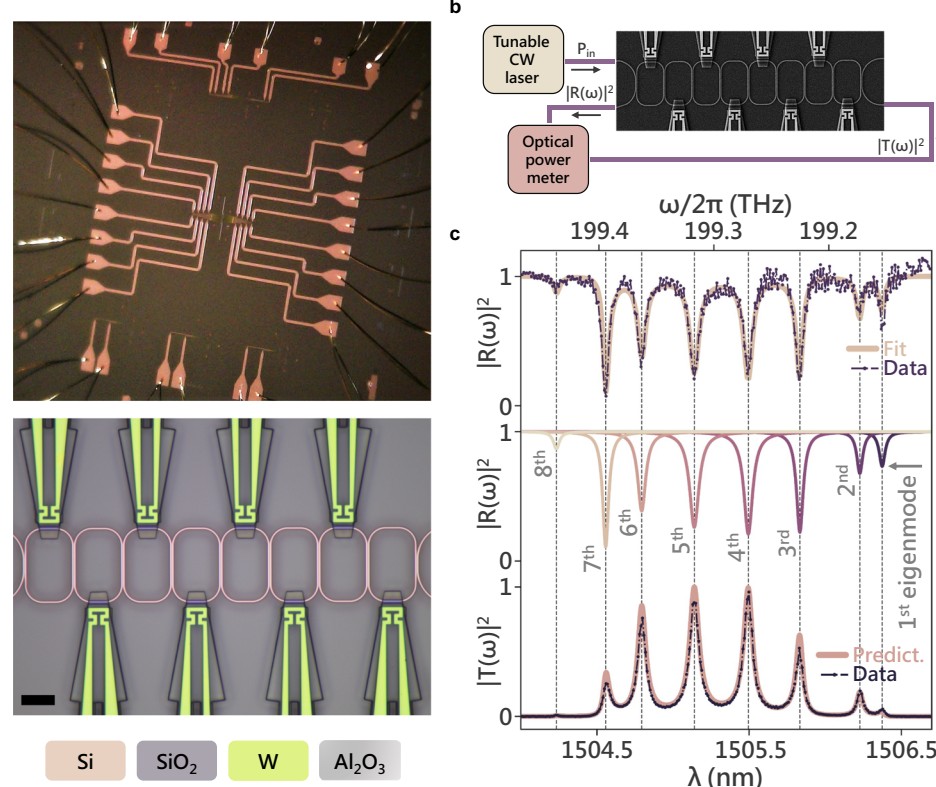

**Fig. 1 | Hamiltonian Tomography. a** Optical image of the electrically controlled CCA depicting the wiring structure, optical micrograph of the CCA (scale bar: $10\,\mu m$). The constituent racetrack resonators are characterized by longer straight segments that are $12\,\mu m$ long, shorter straight segments that are $4\,\mu m$ long, and a bending radius of $5\,\mu m$. **b** Schematic of the experimental setup used for measuring reflection ($|R(\omega)|^2$) and transmission ($|T(\omega)|^2$) spectra. **c** From the top: measured reflection spectrum $|R(\omega)|^2$ (dotted purple) along with the fit generated using the tomography algorithm (cream); followed by a plot showing contributions of various eigenmodes of the system to $|R(\omega)|^2$, and finally at the bottom; experimentally measured transmission spectrum $|T(\omega)|^2$ (dotted purple) along with the predicted transmission spectrum $|T(\omega)|^2$ (pink) from the $H_{eff}^0$ obtained using the tomography algorithm.

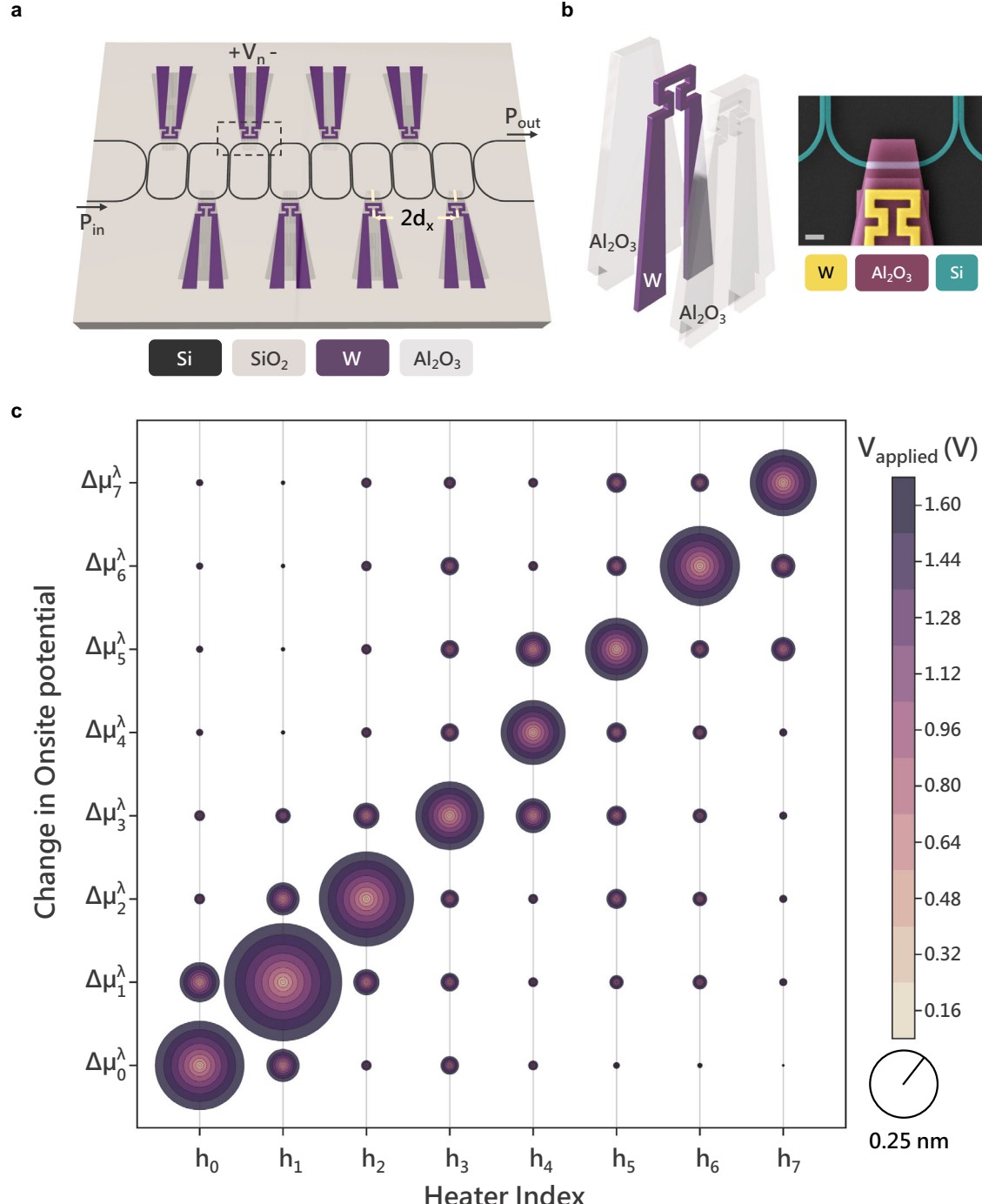

**Fig. 2 | Electrical characterization. a** Device schematic depicting the electrical characterization as voltage $V_n$ is applied to the $n^{th}$ site while measuring the transmission spectrum ($d_x = 14.66\,\mu m$). **b** Exploded view of the TO island heaters. The heater consists of a tungsten element sandwiched between alumina layers. Inset shows a false colored SEM image (scale bar: $2\,\mu m$) of a typical TO island (yellow: tungsten, pink: alumina, teal: silicon). **c** Plot showing the effect of heaters [$h_n$] on the potential profile across the device. The $x$-axis denotes the heater index $h_n$ switched ON for a particular set of measurements and the y-axis represents the change in potential profile [$\Delta\mu_n^\lambda$]. The voltage applied for the measurement ($V_n$) across heater $h_n$ is mapped to the color of the circular surface and the corresponding change in potential is denoted by the radii of the circle encompassing the surface ($0.25nm$ of change is depicted by radii of the circle in the scale bar).

reflection spectrum measurement $\left|R(\omega)\right|^2$ (Fig. 1b) performed at the first site of the CCA by estimating the contribution of individual eigenmodes of the system to the measured spectrum. In Fig. 1c, we plot the reflection spectrum of our CCA along with the corresponding contributions of the 8 individual eigenmodes. We then verify the accuracy of our fit by comparing the experimentally measured transmission spectrum $\left|T(\omega)\right|^2$ of the CCA to the predicted spectrum of the extracted $H_{eff}^0$. Note that, while the reflection spectrum is

needed to map the entire $H_{eff}^0$, the transmission spectrum can be used to find only the eigenvalues of the Hamiltonian (see Supplementary Information: Section S2).

**Design and characterization of the thermal control layer**
Thermal control of the CCA has two primary design objectives: (i) minimizing the additional optical loss incurred when introducing the heaters and (ii) reducing the thermal crosstalk between heaters, which

need to be placed in close proximity owing to the small device footprint necessary to obtain small mode volumes for each cavity and ensure strong coupling between the cavities while maintaining large free spectral ranges[16,19]. We meet both objectives by engineering TO island heaters made up of tungsten ($W$) wires sandwiched between two alumina ($Al_2O_3$) layers (Fig. 2a, b). In such a configuration, the lower thermal resistance of the alumina layers than that of the air/silicon oxide channel separating the islands allows for a more directional transfer of thermal energy from the tungsten heaters to the corresponding resonators. Since alumina is typically optically lossless in the telecommunication wavelength range (see Supplementary Information: Section S2 for ellipsometry data), the islands also allow for placing the tungsten heaters at an adequate distance from the racetrack resonators. This ensures that the introduction of heating elements occurs with minimal absorptive losses and allows for achieving much higher Q-factors required for addressability of individual eigenmodes of a controllable CCA platform. Additionally, the top alumina layer acts as a protective layer against oxidation for the tungsten heating element[20]. In comparison, typical TO control schemes either rely on placing heaters on top of a universal cladding[21] for minimizing additional dissipative losses or incorporating these into the resonator structure itself using photoconductive elements[22] for extremely local control. However, both of these approaches have major drawbacks. While the former allows to obtain very high Q-factors, it suffers from poor local controllability, with our island heaters outperforming these by ∼ 50% in reducing stray effects of thermal crosstalk (see Supplementary Information: Section S3). On the other hand, the latter approach gives extremely local control, but the photoconductive elements inevitably limit the maximum achievable Q-factors due to dopant implantation in the regions that confine the optical mode.

We characterize the realized CCA by applying a linearly increasing voltage across each heater one at a time and recording the respective transmission spectra. The eigenenergies are then extracted from the recorded spectra and combined with our knowledge of $H_{eff}^0$, we estimate the amount of crosstalk between the heaters. The change in the onsite potential $\Delta\mu_n$ when expressed in wavelength units is proportional to the square of voltage $V_n$ applied to the $n^{th}$ site: $\Delta\mu_n^\lambda \propto V_n^2$ (see Supplementary Information: Section S4). To simplify the equations going forward, we express the onsite potentials $\mu_n$ and eigenvalues $\epsilon_n$ of the CCA in wavelength units as $\mu_n^\lambda$ and $\epsilon_n^\lambda$. We plot the effects of voltage $V_n$ applied across heater $h_n$ on the potential profile $[\mu_n^\lambda]$ of the CCA in Fig. 2c. The change in respective onsite potentials $\Delta\mu_n^\lambda$ is represented by the radii of the circles, whereas the color of the circles denotes the voltage $V_n$ applied across heater $h_n$. From the plot, we establish that thermal crosstalk is already low between the nearest neighbors ($n \pm 1$) and becomes negligible as we go beyond the third nearest neighbors ($n \pm 3$).

## CCA control model

We next model the CCA to accurately predict the eigenenergies of the system on application of a voltage profile $[V_n] = [V_0, V_1, \ldots V_{N-1}]$ across the heaters. Here, we define a translationally invariant function $f$ which takes in the input voltage profile and predicts the change in onsite potential when applied at each site. The function $f$ consists of three sets of terms: (i) a fitting correction to the initial onsite potential denoted by $\delta_n$, (ii) thermal contributions from voltages applied across heaters in the thermal neighborhood of site $n$ ($n \pm 3$) connected through proportionality coefficients $\beta_i$'s ($V_i^2$), (iii) cross-terms connected through proportionality coefficients $\gamma_{j,k}$'s ($V_j V_k$ s.t. $i, j, k \in [n-3, n+3]$) accounting for the thermal effects on heater performances by virtue of these being in the thermal vicinity of each other. We also use an additional set of coefficients $\alpha_n$ to incorporate the effects of minor variations in heater resistances due to fabrication

inconsistencies. We then express this relationship mathematically as

$$\Delta\mu_n^\lambda = f([V_n]) = \delta_n + \sum_i \beta_i(\alpha_i V_i^2) + \sum_{j,k} \gamma_{j,k}(\sqrt{\alpha_j \alpha_k} V_j V_k) \quad (3)$$

Note that, we assume that $f$ is translationally invariant, and hence the number of functional parameters $\beta_i$ and $\gamma_{j,k}$ needed to model the device behavior can be restricted to 3 and 12 respectively.

We visualize this process in Fig. 3a where we show how we can use the model to predict the location of eigenenergies $[\epsilon_n^\lambda]$ by finding the eigenvalues of the modified Hamiltonian. Starting with the initial $H_{eff}^0$ and updating its diagonal terms by evaluating the function $f$ at each site of the array for a particular $[V_n]$ we predict the eigenvalues of the modified Hamiltonian as:

$$[\epsilon_n^\lambda]_{predicted} = Eig\left(H_{eff}^0 + [\Delta\mu_n]\mathbb{I}\right) \quad (4)$$

These predicted eigenenergies are then used to fit for $f$ by minimizing the error obtained by calculating the deviations from experimentally extracted eigenenergies across many measurements (here we limit the number of measurements to 288).

$$Error = \frac{||[\epsilon_n^\lambda]_{predicted} - [\epsilon_n^\lambda]_{measured}||^2}{J_{norm}} \quad (5)$$

The probability distribution of the fitting error normalized to the mean hopping-rate $J_{norm}$ is plotted in Fig. 3b. Finally, once we have identified $f$, we use it to predict the location of eigenenergies for 20 randomly generated voltage profiles in Fig. 3c. The centers of the circles in the figure denote the measured values of eigenenergies, and the error in predicted values are represented by the radii of the corresponding circle. The net overall error for a random generation is mapped to the color of the particular set of eigenenergies. From the plot, we can see that the model allows for the prediction of the eigenenergies of our system with greater than 96% accuracy.

## Discussion

To summarize, we demonstrated a thermally controlled optical CCA which can be used to realize a set of tight-binding Hamiltonians with addressability to the entire quantized eigenenergy spectrum. To ensure a compact device size and high-Q cavities necessary (albeit not sufficient) to reach the regime of interacting photons and allow access to the full quantized eigen-spectrum[12], we engineered special TO islands heaters, which reduced the effects of thermal crosstalk by almost 50% over previously reported works[21,23] and allowed Q-factors up to $7.1 \times 10^4$ for heater integrated racetrack resonators. Finally, we presented a mathematical model which allowed for precise control of the eigenenergies of the implemented Hamiltonians within an error of only 4% of the mean hopping rate. Our device can already be used to simulate a number of single-particle physical effects like Anderson localization[13] and the Su–Schrieffer–Heeger (SSH) model[24]. One potential disadvantage of using TO heaters is that our dynamic modulation rates are limited to the *MHz* regime[21] which rules out the possibility to implement models like the Haldane quantum Hall effect[25] requiring modulation of onsite potentials at rates comparable to the mean hopping rate (∼ *GHz*). However, what TO heaters might lack, they make up for it by allowing a larger range of static modulation and ease of scalability in comparison to say electro-optical modulators which might be much faster but present far more challenges when it comes to scaling to a larger number of sites[26]. Looking ahead, our TO island

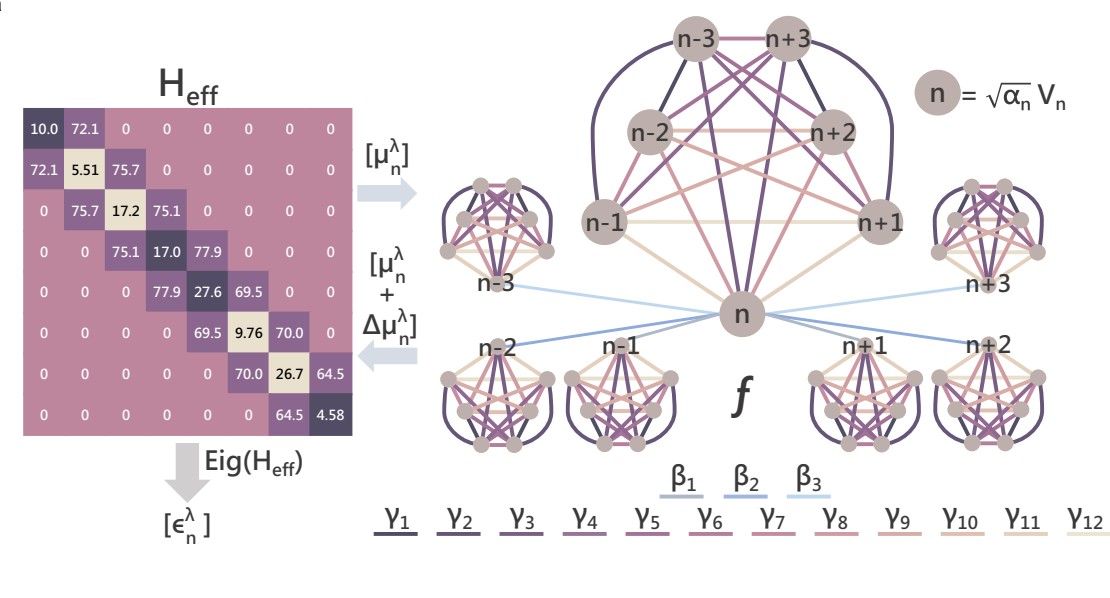

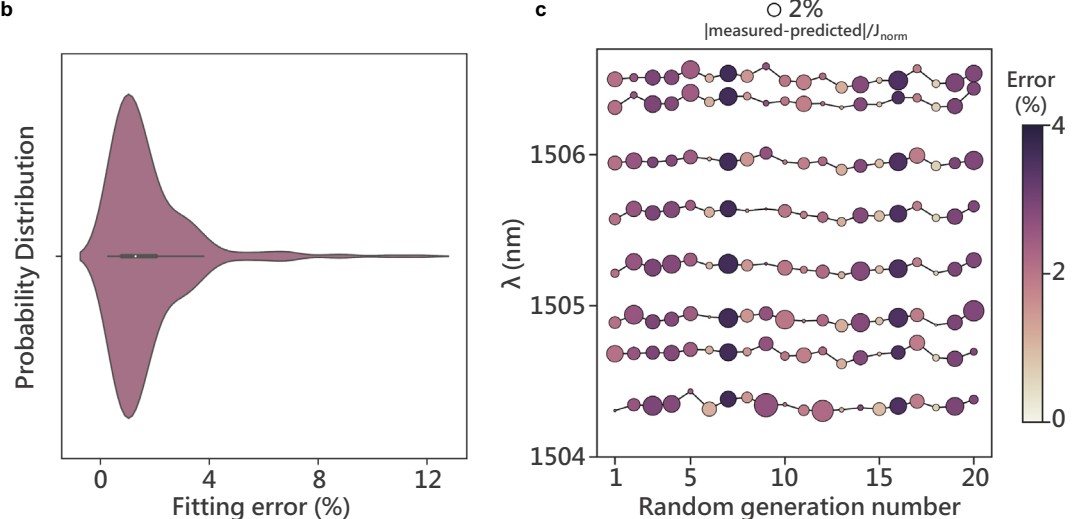

**Fig. 3 | Electrical control model and eigen-energy prediction. a** Visualization of the optimization process depicting how the model takes in the system Hamiltonian $H_{eff}$ and fits for the function $f$ which connects the applied voltage profile $[V_n]$ to change in onsite potentials. We predict the position of eigen-energies on application of $[V_n]$ by calculating the change in onsite potentials which lie along the Hamiltonian diagonal and finding the eigenvalues of the modified Hamiltonian. The optimization is initialized using $H^0_{eff}$ shown in the matrix form (only real part is depicted). All entries are in $GHz$, with diagonal terms denoting the deviations in resonant frequency about the mean (dark purple: $+ve$ deviation, tan: $-ve$ deviation) and super/sub diagonal terms denoting the hopping rates. $n \pm i$ denote indices of sites in neighborhood of site $n$ on which the function $f$ is being applied on. The coefficients $\beta_i$ are denoted in hues of blue and $\gamma_{j,k}$ are denoted in hues of purple. **b** Violin plot denoting the fitting error normalized to the mean hopping rate $J_{norm}$ across 288 points. **c** Prediction accuracy plot where the x-axis denotes the random generation, and the y-axis denotes the wavelength. The location of the measured eigen-energies is denoted by the dark black lines in background. The radii of the circles denote the deviation of the predicted value from measured values (scale on top). The color of the dots denotes the overall prediction error for that generation.

heaters equipped CCAs are hence perfectly suited for implementing a large class of Hamiltonians that do not require very high-speed modulation, such as the Hofstadter Hamiltonian[3,19], SSH Hamiltonian[16,24] and non-Hermitian topological Hamiltonians[27], among others. Further, leveraging the immense scaling potential of photonics, operating in a linear regime the current CCA can be scaled to sizes where it can be used to study classical and quantum bosonic walks and solid-state lattice band structures[28]. While we did not demonstrate any non-linearity, our CCA with its small footprint, high Q-factors and a cladding free design has the potential to enable integration with excitonic materials[29], defect centers[30,31] and possibly reach single photon non-linear regime[32–34]. Additionally, adopting more complex control algorithms involving feedback control[35,36] and data-based learning[37] in future works can help to improve the accuracy of the realized Hamiltonians further. In conclusion, our

work shows the scalability, programmability, and measurability of photonic CCAs for the first time, and is a significant step forward over state-of-the-art photonic quantum simulators, which are traditionally neither programmable nor tomographically mappable.

## Methods

### Design
Ansys Lumerical FDTD, MODE and HEAT were used to simulate and optimize the device parameters.

### Fabrication
A silicon on insulator wafer (SOITEC) with $220nm$ thick film of silicon on $3\mu m$ thick buried silicon oxide was diced. A $10\,mm \times 10\,mm$ chip thus obtained was used for further processing. After cleaning, the chip was spin-coated with Hydrogen

silsesquioxane (*HSQ*) and exposed using a JEOL JBX6300FS electron beam (e-beam) lithography system. After developing in 25% *TMAH*, the chip was etched using an inductively coupled plasma etcher with a $Cl_2$ chemistry. The resist was then removed using diluted *BOE*. The chip then underwent several cycles of patterning, followed by electron beam evaporation/sputtering of materials and lift-off to define the island heaters, and contact pads (see Supplementary Information: Methods for the fabrication flow diagram). The first of these cycles involved defining the island pattern in positive tone polymethyl methacrylate (*PMMA*) resist using e-beam lithography, followed by depositing 265*nm* thick $Al_2O_3$ layer using evaporation and finally lift-off to obtain the lower layer of the islands. The next cycle began by patterning of heating elements using a similar *PMMA* based e-beam lithography step. A 150*nm* thick tungsten (*W*) layer was then sputtered, followed by a sonication-based lift-off to obtain the heaters. The contact pads made up of 25*nm Ti*/325*nm Pt* layers were then defined using a *PMMA* based e-beam lithography followed by an evaporation and lift-off cycle. The final 300*nm* thick $Al_2O_3$ cladding over the islands was then obtained using a similar e-beam lithography/evaporation/lift-off process cycle.

### Measurement setup

The spectrum of the fabricated device was measured via a fiber coupled setup in which the input light was provided by a tunable continuous-wave laser (Santec TSL-510) and a low-noise power meter (Keysight 81634B) was used to collect the output light from the grating couplers. A DAQ (MCC USB 3114) was used to apply the electrical potential profile across the device.

## Data availability

Additional data related to this paper may be requested from the authors on request.

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

## Acknowledgements

This work was funded through the National Science Foundation grants NSF-QII-TAQS-1936100, NSF-1845009. Part of this work was conducted at the Washington Nanofabrication Facility/Molecular Analysis Facility, a National Nanotechnology Coordinated Infrastructure (NNCI) site at the University of Washington with partial support from the National Science Foundation via awards NNCI-1542101 and NNCI-2025489. We thank J. Simon for pointing us towards the Hamiltonian tomography algorithms; M. Zhelyeznyakov and S.L. Brunton for help in numerical optimization; J. Whitehead for help in automating the measurements; Y. Chen for help in fabrication in very initial stages of the project.

## Author contributions

A.S. and A. Majumdar conceptualized the idea. A.S. designed the device and fabricated it. A.S. then programmed the laser, power meter and the electrical control unit for the experiment. The device was then characterized by A.S. and A. Manna. A.S., A. Manna and R.T. came up with a modification to the algorithm used for Hamiltonian tomography. A.S., A. Manna and A. Majumdar then analyzed the electrical control data. A.S. and A. Majumdar drafted the initial manuscript. All the authors contributed to the interpretation of the results and to the writing of the manuscript. A. Majumdar supervised the project.

## Competing interests

The authors declare no competing interests.
