## [Peer review file · Nature Communications]

Reviewers' comments:

Reviewer #1 (Remarks to the Author):

The authors construct a set of eight linearly coupled optical cavities, add fibre couplers to the first and the last of these resonators to provide access to reflection spectra of this system. Probing the system with classical light then allows the authors to reverse engineer the Hamiltonian parameters of the system. In the discussion in order to enhance the perception of importance of their work the authors then state that it "will readily allow integration with excitonic materials and possibly reaching single photon non-linearities". I am afraid that such a step is highly unlikely to be "readily" available and more than a statement of belief, i.e. a careful theoretical analysis, will be needed to provide a basis for such an optimistic statement.

It should also be noted that Hamiltonians of the type that have been realised here have already been achieved in a wide variety of physical systems. These include trapped ions where the spin motion coupling provides a handle to couple different mechanical oscillations, in nanomechanical networks (PRB 101, 174303 (2020)) both of which are clearly reconfigurable. Furthermore, I would like to stress that Nature 566, 51 (2019) has demonstrated coupled cavity arrays based on superconducting qubits that realise a Bose-Hubbard Hamiltonian thus going far beyond what has been achieved in this work by introducing non-linearities. Given the lack of detail in the discussion it is by no means clear what advantage the "readily" available system that the authors are alluding to would have over these and many other experimental realisations.

While the present work may be published in some form it is certainly not of sufficient importance to be considered for publication in Nature Communications.

Coupled cavity arrays have been realised in a variety of technologies, e.g.

Reviewer #2 (Remarks to the Author):

Report on Saxena et al.

The manuscript reports on an experimental realization of a coupled-cavity system as a photonic simulator. As far as I can tell, one of the key achievements is the addressability of individual sites via thermo-optic island heaters. The authors demonstrate that this works well, i.e. crosstalk between sites is rather small. As the central result, the authors show that by using a theoretical model and a fitting procedure, they can obtain parameters for onsite- and crosstalk contributions and, thereby, uniquely determine the physical implementation of the photonic simulator. Once the fitting procedure is successfully done, the eigenspectrum of randomly generated tight-binding Hamiltonians is successfully predicted by the "trained" model as shown by comparison with measurements.

I must say that I find the manuscript hard to follow and many questions remain open. For this reason alone, I cannot recommend publication in its current form, and I will give a list of suggestions for improvements below. At the same time, I would like to emphasize that as a theoretical physicist, I am not in a position to judge the quality and novelty of the actual experimental implementation of the photonic simulator. I do believe that the work is impressive and that a system that can simulate a variety of 8-site tight-binding Hamiltonians is useful for many things. However, not a single application is named in the text.

In its current form, the article is rather written for an applied journal. If the authors wish to address a broader audience, the text should be more illustrative and motivation and prospects should be made

clear to those not working in exactly the same field.

I suggest the following:

- The paper is quite short and ends very abruptly. The conclusion should indicate what the achievement means. For a broad interest journal, this is even more important: Why should one be excited about this?
- The paper is written either from a technical or a theoretical perspective. There is no overarching scope or red thread with a physical picture in mind - this makes the text hard to follow. Is the main achievement the realization of site-controllability with the island heaters? If so, what is state-of-the-art, how much of an improvement is this here? If the main achievement is the demonstration of the TB realizations, then what is one going to do with it? How useful are 8 sites for practical applications?
- The results in Fig. 3 are hard to discern. What exactly can be the purpose of Fig. 3a (right panel) - what are the $n-1$, $n-2$, $n+1$, ...? There is hardly any discussion in the text.
- What is the advantage of CCAs with emitters over purely cavity-based ones?
- The work focuses purely on the spectrum. Do dynamical effects play any role?
- In how far is dissipation a limiting factor here? (There are several contributions in Eq. 2)

We are very grateful to the referees for reading the manuscript and providing many insightful comments. Our point-to-point responses to the referee's comments appear below.

Reviewer #1 (Remarks to the Author):

The authors construct a set of eight linearly coupled optical cavities, add fibre couplers to the first and the last of these resonators to provide access to reflection spectra of this system. Probing the system with classical light then allows the authors to reverse engineer the Hamiltonian parameters of the system. In the discussion in order to enhance the perception of importance of their work the authors then state that it "will readily allow integration with excitonic materials and possibly reaching single photon non-linearities". I am afraid that such a step is highly unlikely to be "readily" available and more than a statement of belief, i.e. a careful theoretical analysis, will be needed to provide a basis for such an optimistic statement.

Response: In the last line of our paper, we wrote: “will readily allow integration with excitonic materials and possibly reaching single photon non-linearities” and additionally cited a previous work from our group to support the statement of such integration. The term readily here was present to highlight the ease with which such integration had already been performed by many groups previously. We also have done several theoretical and experimental works since, highlighting the path to reach single photon linearity using such excitonic materials.

While we agree with the reviewer that reaching that regime is not “readily” possible it is an outstanding problem that an entire photonics community is actively working on. We had stated in the previous line in the manuscript that tackling this problem is not the purpose for this paper: “While we did not demonstrate any non-linearity, our CCA with high Q-factors and a cladding free design” where we highlight also the two conditions that a photonic CCA must meet to allow for this ready integration and how our work is the first one to do so.

We have edited the last paragraph to address the issue in terminology and added additional references to highlight the pathway towards reaching the regime as follows:

“While we did not demonstrate any non-linearity, our CCA with high Q-factors and a cladding free design will potentially allow integration with excitonic materials²⁴ and possibly reaching single photon non-linearities^{25–27}. Nevertheless, our work shows the scalability, programmability, and measurability of photonic CCAs for the first time, and a significant step forward over state-of-the-art photonic quantum simulators, which are traditionally not programmable or tomographically mappable.”

It should also be noted that Hamiltonians of the type that have been realised here have already been achieved in a wide variety of physical systems. These include trapped ions where the spin motion coupling provides a handle to couple different mechanical oscillations, in nanomechanical networks (PRB 101, 174303 (2020)) both of which are clearly reconfigurable. Furthermore, I would like to stress that Nature 566, 51 (2019) has demonstrated coupled cavity arrays based on

superconducting qubits that realise a Bose-Hubbard Hamiltonian thus going far beyond what has been achieved in this work by introducing non-linearities. Given the lack of detail in the discussion it is by no means clear what advantage the "readily" available system that the authors are alluding to would have over these and many other experimental realisations.

Response: We agree with the reviewer that superconducting qubits have been used to realize many a Hamiltonians and are the front running platform for analog simulations. We had also mentioned this in the introduction in the paper as: "While there have been numerous works on analog quantum simulation with microwave photons (3–6)" and have cited similar works to support this.

We have also now included highlighted work to further back this claim.

However, we had also followed this up with why realizing such systems in photonics is advantageous. Citing from the paper again: "higher energy optical photons can provide several additional advantages, including operability at much higher temperatures (7), which would significantly simplify the experiments and lower the resources needed to scale the simulator; and availability of single photon detectors, which would allow direct measurement of multiparticle correlations (8, 9)".

Elaborating further, superconducting systems have been remarkably successful in demonstrating prototypical quantum simulators with a few 10s of sites. However, in order to realize quantum simulators that can simulate actual quantum materials, we need to scale to a lot larger number of sites (in thousands). Such a scaling is fundamentally hard to do in superconducting systems owing to limitation of both fabrication techniques and experimental challenges of operating at near 0K temperatures. This is where photonics is extremely useful, however, to realize quantum simulators in photonics requires realization of CCAs that satisfy four broad criteria:

1. scalability (how to scale photonic CCAs in a way to overcome effects of nanofabrication disorders),
2. programmability (how to control optical cavities while ensuring minimal additional losses),
3. measurability (how to identify underlining Hamiltonian with limited access) and
4. non-linearity (access to non-linearity).

In our paper, we had tried to convey from the introduction itself that we were tackling the first three challenges in our current manuscript, building on our previous work (ACS Photonics 9.2 (2022): 682-687). In photonics achieving programmability and measurability while ensuring low dissipation and disorder is fundamentally harder to achieve and we believe ours is the first work demonstrating it in a photonic coupled cavity system. The fourth challenge has been a topic of investigation in other works and is still an active topic of research.

The other citation (PRB 101, 174303 (2020)) and ion spins refer to experimental systems which are completely different from photonics. They have their own fundamental challenges, and while they have demonstrated some of the four conditions needed to realize quantum simulators, we re-iterate that realizing such systems in CMOS compatible photonics is still one of the most promising

ways forward to realize truly useful quantum simulators and our work is an impactful step towards it. As such, given that there is no clear front-running physical system for quantum simulation, demonstrating similar physics in a new system is important. Hence, we respectfully disagree with the reviewer about this aspect.

We have edited the paper to include this discussion in the introduction as follows:

“ While there have been numerous works on analog quantum simulation with microwave photons³⁻⁷, optical photons with their extremely weak interaction with environment can provide several additional advantages. The higher energy of optical photons allows for preservation of quantum states even at room temperature and hence enabling operability at much higher temperatures⁸, which significantly simplifies the experiments and lowers the resources needed to scale the simulator. Additionally, availability of single photon detectors in optical domain allows direct measurement of multiparticle correlations^{9,10} which are a key set of measurements for characterizing the quantum-ness of realized states. Building on recent advancements in nanofabrication, quantum optical systems have shifted from bulky tabletop systems prone to misalignments to fully integrated on chip photonic circuits. These large scale photonic integrated circuits owing to their small size and high speed of operation present opportunities for unprecedented scalability to “practical quantum advantage”¹¹.

One solution to engineer such quantum systems in optics is via photonic coupled cavity arrays (CCA)¹² where coupling between cavities provides a potential map for photons to move around, and strong spatial confinement of light for long durations allows access to onsite non-linearity via coupling with various excitonic materials. For photonic CCAs to be used as quantum simulators four broad requirements need to be satisfied, namely, (i) scalability: there must exist pathways to scale to a large number of sites; (ii) measurability: there is a need for protocols to perform Hamiltonian tomography with restricted access and have CCAs with addressability to all the eigenstates of the system; (iii) controllability: control over all the terms describing the Hamiltonian is required; and finally (iv) optical non-linearity: need to realize photon-photon interaction to simulate many-body Hamiltonians. The last demand as a precondition requires using high-quality factor (Q) cavities with small mode volumes as constituents of the CCA. Such high-Q cavities are also necessary to probe the entire quantized eigenenergy spectra. Though several experiments showing various physical phenomena using optical CCAs have been previously reported¹³⁻¹⁵, none of these CCAs are programmable and can have access to the entire quantized eigen-energy spectra of the Hamiltonian. While careful selection of operation regime can lead to a pathway that allows scalability to multiples sites using photonics¹⁶, in the optical regime achieving programmability and measurability of the eigen-spectrum, is very challenging owing to the extremely small physical dimensions involved. ”

While the present work may be published in some form it is certainly not of sufficient importance to be considered for publication in Nature Communications.

Coupled cavity arrays have been realised in a variety of technologies, e.g.

Response: It seems that the reviewer response is truncated for some reason on the portal. But we believe this concern goes back to the previous comment of the reviewer. Indeed, many different platforms have been used to implement coupled cavity architecture. However, demonstrations with high-Q cavities in photonics are missing. We argued how photonics could be beneficial over other physical systems. Again, given that there is no clear front-running physical system for quantum simulation, demonstrating similar physics in a new system is important. Hence, we respectfully disagree with the reviewer about this aspect.

We have made the changes indicated above to highlight this discussion.

We thank the reviewer for raising valuable comments and hope that we have addressed their concern in our revised manuscript.

Reviewer #2 (Remarks to the Author):

We are grateful to the referee for carefully reading the manuscript and providing many insightful comments. Our point-to-point responses to the referee's comments appear below.

The manuscript reports on an experimental realization of a coupled-cavity system as a photonic simulator. As far as I can tell, one of the key achievements is the addressability of individual sites via thermo-optic island heaters. The authors demonstrate that this works well, i.e. crosstalk between sites is rather small. As the central result, the authors show that by using a theoretical model and a fitting procedure, they can obtain parameters for onsite- and crosstalk contributions and, thereby, uniquely determine the physical implementation of the photonic simulator. Once the fitting procedure is successfully done, the eigenspectrum of randomly generated tight-binding Hamiltonians is successfully predicted by the "trained" model as shown by comparison with measurements.

I must say that I find the manuscript hard to follow and many questions remain open. For this reason alone, I cannot recommend publication in its current form, and I will give a list of suggestions for improvements below. At the same time, I would like to emphasize that as a theoretical physicist, I am not in a position to judge the quality and novelty of the actual experimental implementation of the photonic simulator. I do believe that the work is impressive and that a system that can simulate a variety of 8-site tight-binding Hamiltonians is useful for many things. However, not a single application is named in the text.

In its current form, the article is rather written for an applied journal. If the authors wish to address a broader audience, the text should be more illustrative and motivation and prospects should be made clear to those not working in exactly the same field.

Response: We thank the reviewer for appreciating the significance of achieving an 8-site tight binding photonic simulator. Due to the experimental nature of the work, we expounded on the engineering challenges we needed to overcome to realize such a system.

In broader terms, superconducting systems have succeeded in demonstrating prototypical quantum simulators with few 10s of sites. However, in order to realize quantum simulators that can simulate

actual quantum materials, we need to scale to a lot large number of sites (in thousands or more). Such a scaling is fundamentally hard to do in superconducting systems owing to limitation of both fabrication techniques and experimental challenges of operating at near $0K$ temperatures. This is where photonics is extremely useful as we can leverage the advances made in CMOS fabrication and operate the devices at room temperature. Additionally, photonics also allows access to single photon detectors allowing direct measurements of multi-photon correlations which is not possible to do in superconducting platform.

However, realization of photonic analog quantum simulators depends on realization of photonic CCAs that satisfy the following criteria of:

1. scalability (how to scale photonic CCAs in a way to overcome effects of nanofabrication disorders),
2. programmability (how to control optical cavities while ensuring minimal additional losses),
3. measurability (how to identify underlining Hamiltonian with limited access) and
4. non-linearity (access to non-linearity).

In our work we successfully demonstrated a way to tackle the first three of the challenges faced in realizing photonic quantum simulators.

We have updated the introduction of the paper to include this discussion as follows:

“Achieving analog quantum simulation necessitates the realization of programmable quantum devices¹. Due to their inherent driven-dissipative nature, photonic systems are a promising platform for non-equilibrium quantum simulation². An archetypal photonic quantum simulator consists of an array of programmable non-linear nodes with access to the entire quantized eigen-energy spectra of the Hamiltonians being simulated. While there have been numerous works on analog quantum simulation with microwave photons³⁻⁷, optical photons with their extremely weak interaction with environment can provide several additional advantages. The higher energy of optical photons allows for preservation of quantum states even at room temperature and hence enabling operability at much higher temperatures⁸, which significantly simplifies the experiments and lowers the resources needed to scale the simulator. Additionally, availability of single photon detectors in optical domain allows direct measurement of multiparticle correlations^{9,10} which are a key set of measurements for characterizing the quantum-ness of realized states. Building on recent advancements in nanofabrication, quantum optical systems have shifted from bulky tabletop systems prone to misalignments to fully integrated on chip photonic circuits. These large scale photonic integrated circuits owing to their small size and high speed of operation present opportunities for unprecedented scalability to “practical quantum advantage”¹¹.

One solution to engineer such quantum systems in optics is via photonic coupled cavity arrays (CCA)¹² where coupling between cavities provides a potential map for photons to move around, and strong spatial confinement of light for long durations allows access to onsite non-linearity via coupling with various excitonic materials. For photonic CCAs to be used as quantum simulators four broad requirements need to be satisfied, namely, (i) scalability: there must exist pathways to scale to a large number of sites; (ii) measurability: there is a need for protocols to perform

Hamiltonian tomography with restricted access and have CCAs with addressability to all the eigenstates of the system; (iii) controllability: control over all the terms describing the Hamiltonian is required; and finally (iv) optical non-linearity: need to realize photon-photon interaction to simulate many-body Hamiltonians. The last demand as a precondition requires using high-quality factor (Q) cavities with small mode volumes as constituents of the CCA. Such high-Q cavities are also necessary to probe the entire quantized eigenenergy spectra. Though several experiments showing various physical phenomena using optical CCAs have been previously reported^{13–15}, none of these CCAs are programmable and can have access to the entire quantized eigen-energy spectra of the Hamiltonian. While careful selection of operation regime can lead to a pathway that allows scalability to multiples sites using photonics¹⁶, in the optical regime achieving programmability and measurability of the eigen-spectrum, is very challenging owing to the extremely small physical dimensions involved. ”

I suggest the following:

- The paper is quite short and ends very abruptly. The conclusion should indicate what the achievement means. For a broad interest journal, this is even more important: Why should one be excited about this?

We addressed the length of the paper by discussing the motivation behind the paper in detail in the introduction highlighted above. We have also expanded the conclusion to indicate the potential uses of our device and its broader significance. The edited conclusion is as follows:

“In this work, we demonstrated a thermally controlled optical CCA which can be used to realize a set of tight binding Hamiltonians with addressability to the entire quantized eigen-energy spectrum. To ensure a compact device footprint and high Q cavities necessary for reaching the regime of interacting photons and allowing access to the full quantized eigen-spectrum¹², we engineered special TO islands heaters, which allowed reduction in thermal crosstalk by almost 50% over previously reported works^{19,21}. Our device design allowed us to realize Q factors more than 8.5×10^4 for heater integrated racetrack resonators. Finally, we presented a mathematical model which allowed for precise control of the eigen-energies of the implemented Hamiltonians within an error of only 4% of the mean hopping rate.

Our device can already be used to simulate a number of single-particle physical effects like Anderson localization¹³ and the Su–Schrieffer–Heeger model²². Further, leveraging the immense scaling potential of photonics, operating in a linear regime the CCA can be scaled to sizes where it can be used to study classical and quantum bosonic walks and solid-state lattice band structures²³. While we did not demonstrate any non-linearity, our CCA with high Q-factors and a cladding free design will potentially allow integration with excitonic materials²⁴ and possibly reaching single photon non-linearities^{25–27}. Nevertheless, our work shows the scalability, programmability, and measurability of photonic CCAs for the first time, and a significant step forward over state-of-the-art photonic quantum simulators, which are traditionally not programmable or tomographically mappable.”

- The paper is written either from a technical or a theoretical perspective. There is no overarching scope or red thread with a physical picture in mind - this makes the text hard to follow. Is the main achievement the realization of site-controllability with the island heaters? If so, what is state-of-the-art, how much of an improvement is this here? If the main achievement is the demonstration of the TB realizations, then what is one going to do with it? How useful are 8 sites for practical applications?

Response: The realization of site control is indeed one of the main achievements of our work. We have included a detailed comparison of how our design is at least 50% better than the current state-of-the-art heaters in the supplementary information. For the ease of the reviewer, we are also including it below.

“For a racetrack resonator we know that $n_{\text{eff}}l = m\mu_n^\lambda$, where n_{eff} is the refractive index of the racetrack resonator, l is the length of the resonator, μ_n^λ is the onsite potential (in wavelength units) and $m \in \mathbb{Z}$. Let a segment of length x be affected by change in temperature, such that its refractive index is given by $n(x)$. Resonance condition for the resonator then becomes:

$$n_{\text{eff}}(l - x) + \int n(x)dx = m(\mu_n^\lambda + \Delta\mu_n^\lambda) \quad (\text{S29})$$

Removing the constant terms gives

$$\int (n(x) - n_{\text{eff}})dx = \int \Delta(n(x))dx = m(\Delta\mu_n^\lambda) \quad (\text{S30})$$

Assuming a constant thermo-optic coefficient dn/dT we can write $\Delta n(x) = \rho\Delta T(x)$. Consequently,

$$\int \rho\Delta T(x)dx = m(\Delta\mu_n^\lambda) \quad (\text{S31})$$

This implies shift in onsite potential $\Delta\mu_n^\lambda \propto \int \Delta T(x)dx$.

Next; to estimate this effect of thermal crosstalk in our system and compare it with typically used thermo-optic (TO) heaters we perform a set of thermal simulations using ANSYS Lumerical HEAT (Fig. S3). In Fig. S3A we have the schematic depicting a conventional TO heater, where the metallic heating element sits directly on top of the resonator segment separated by a uniform and universal $1.5\mu\text{m}$ SiO_2 cladding. In Fig. S3B we have the island TO heater design we used for our device. For both these cases we have the CCA made up of only 3 sites with the heater placed on the middle site (n^{th}). We then record the temperature profile in the shorter straight segment (highlighted in yellow) of the racetrack resonators for the n^{th} and $(n + 1)^{\text{th}}$ sites as we vary the voltage applied V_n across the heater from 0V to 0.8V. In Fig. S3C we plot the average temperature across the segments given by

$$\text{Avg}(T) = \frac{\int T(x)dx}{\int dx} \quad (\text{S32})$$

for neighboring $(n + 1)^{\text{th}}$ site. Similarly, in Fig. S3D we plot the average temperature across the segment for n^{th} site. It is clear from these plots that even though both heater designs cause a similar increase in the onsite temperature (n^{th}), the average temperature in the neighboring sites diverges rapidly between the designs with the temperature difference already greater than 30K at 0.8V. To study the difference in effects of this thermal crosstalk we then define a dimensionless parameter η given by

$$\eta = \frac{\Delta\mu_{n+1}^\lambda}{\Delta\mu_n^\lambda} \quad (\text{S33})$$

Using Eq. S31 we can write the above as

$$\eta = \frac{\Delta\mu_{n+1}^\lambda}{\Delta\mu_n^\lambda} = \frac{\int \Delta T_{n+1}(x)dx}{\int \Delta T_n(x)dx} = \frac{\int (T_{n+1}(x) - 300)dx}{\int (T_n(x) - 300)dx} \quad (\text{S34})$$

In Fig. S3E we plot the η (as %) for both the designs. We can see from the plot that $\eta_{\text{cladded}} \sim 0.056$ and $\eta_{\text{island}} \sim 0.024$. This implies that the islands TO heaters outperform the typical TO heaters in reducing the effects of thermal crosstalk in the device by more than 50%. In general, we do not expect the η values to vary with the voltage V_n (see Section S3) as also verified by the plots.

Fig. S3. Thermal simulation results. (A) Schematic depicting the device with conventional TO heater sitting directly over the resonator separated by 1.5 μm thick oxide cladding (label: Cladded). (B) Schematic depicting a device equipped with the island TO heater used in our paper (label: Island). (C) Plot comparing average temperature across the straight segment (in yellow) of the (n + 1)th resonator. (D) Plot comparing average temperature across the straight segment (in yellow) of the nth resonator. (E) Plot comparing η factor in % for the two device designs. (purple: cladded, pink: island)”

We have also edited the manuscript to include a more detailed discussion on this as:

“Thermal control of the CCA has are two primary design objectives: (i) minimizing the additional optical loss incurred when introducing the heaters, and (ii) reducing the thermal crosstalk between heaters which need to be placed in close proximity owing to the small device footprint necessary to obtain small mode volumes for each cavity and ensure strong coupling between the cavities¹⁶. We meet both objectives by engineering TO island heaters made up of tungsten (W) wires sandwiched between two alumina (Al₂O₃) layers (Fig. 2a, b). In such a configuration the lower thermal resistance of the alumina layers than that of the air/silicon oxide channel separating the islands allows for a more directional transfer of thermal energy from the tungsten heaters to the corresponding resonators. Since alumina is optically lossless in the telecommunication wavelength

range, the islands also allow for placing the tungsten heaters at an adequate distance from the racetrack resonators. This ensures that the introduction of heating elements occurs with minimal absorptive losses and allows for achieving much higher Q-factors required for addressability of individual eigenmodes of a controllable CCA platform. In comparison, typical TO control schemes either rely on placing heaters on top of a universal cladding¹⁹ for minimizing additional dissipative losses or incorporating these into the resonator structure itself using photoconductive elements²⁰ for extremely local control. However, both these approaches have major drawbacks. While the former allows to obtain very high Q factors it suffers from poor local controllability, with our island heaters outperforming these by more than 50% in reducing stray effects of thermal crosstalk (see supplementary information). On the other hand, while the latter approach gives extremely local control, the photoconductive elements used to do so inevitably limit the maximum achievable Q-factors due to dopant implantation in the regions which host the optical mode.”

Demonstration of tight binding Hamiltonians is the other result from our paper, and we have expanded on its potential uses in the conclusion of the paper as:

“ Our device can already be used to simulate a number of single-particle physical effects like Anderson localization¹³ and the Su–Schrieffer–Heeger model²². Further, leveraging the immense scaling potential of photonics, operating in a linear regime the CCA can be scaled to sizes where it can be used to study classical and quantum bosonic walks and solid-state lattice band structures²³. While we did not demonstrate any non-linearity, our CCA with high Q-factors and a cladding free design will potentially allow integration with excitonic materials²⁴ and possibly reaching single photon non-linearities^{25–27}. Nevertheless, our work shows the scalability, programmability, and measurability of photonic CCAs for the first time, and a significant step forward over state-of-the-art photonic quantum simulators, which are traditionally not programmable or tomographically mappable.”

- The results in Fig. 3 are hard to discern. What exactly can be the purpose of Fig. 3a (right panel) - what are the $n-1, n-2, n+1, \dots$? There is hardly any discussion in the text.

Response: The right panel in Fig. 3a is present as an aid for ease of visualization of how the function f models the change in onsite potential when a voltage profile is applied across the array. The $n - 1, n - 2, n + 1$ etc. denote the neighborhood of site n on which the function f is being evaluated to estimate its change in onsite potential denoted by $\Delta\mu_n^\lambda$.

We have included a discussion in the paper explaining the definition of function f . It is as follows:

“We next model the CCA to accurately predict the eigenenergies of the system on application of a voltage profile $[V_n] = [V_0, V_1, \dots, V_{N-1}]$ across the heaters. To do so, we define a translationally invariant function f which takes in the input voltage profile and predicts the change in onsite potential when applied at each site. The function f consists of three sets of terms: (i) a fitting correction to the initial onsite potential denoted by δ_n , (ii) thermal contributions from voltages

applied across heaters in the thermal neighborhood of site n ($n \pm 3$) connected through proportionality coefficients β_i 's (V_i^2), (iii) cross-terms connected through proportionality coefficients $\gamma_{j,k}$'s ($V_j V_k$ s.t. $i, j, k \in [n - 3, n + 3]$) accounting for the thermal effects on heater performances by virtue of these being in the thermal vicinity of each other. We also use an additional set of coefficients α_n to incorporate the effects of minor variations in heater resistances due to fabrication inconsistencies.”

We have also included the following line in the figure description:

“ $n \pm i$ denote indices of sites in neighborhood of site n on which the function f is being applied on.”

- What is the advantage of CCAs with emitters over purely cavity-based ones?

Response: Achieving strong nonlinearity from bulk optical materials has proved to be an extremely hard problem. For examples, researchers have explored the possibility of using bulk nonlinearity to reach quantum regime, but the requirement of Q-factor is $\sim 10^6 - 10^7$, making it very difficult. (Ferretti, Sara, and Dario Gerace, Physical Review B 85.3 (2012): 033303; Majumdar, Arka, and Dario Gerace. Physical Review B 87.23 (2013): 235319.). The community has hence focused on integrating optical materials with either color defects or excitonic materials to reach a regime for attaining single photon optical nonlinearity. For example, excitonic materials owing to their huge dipole moments and electron-electron interactions can lead to strong optical nonlinearities (eg. in "Highly nonlinear dipolar exciton-polaritons in bilayer MoS2." Nature communications 13.1 (2022). Hence, for our work we specifically designed a CCA which in future will be compatible for integration with all sorts of potential emitters.

- The work focuses purely on the spectrum. Do dynamical effects play any role?

Response: The spectrum was the focus of our work because we wanted to realize an experimental setup which allowed characterization of the underlying Hamiltonian using a steady state measurement procedure. If dynamical characterization is required for probing a particular effect in mind, our device design is fully compatible to be used for that as well. However, for the purposes of Hamiltonian tomography we show that it possible to do so completely with the more accessible steady state characterization.

- In how far is dissipation a limiting factor here? (There are several contributions in Eq. 2)

Response: Dissipation in quantum systems is a leading source of decoherence. In principle we typically want the system to have as low a dissipation rate as possible, however integrating optical control elements without increasing the dissipation is extremely hard to do experimentally. Despite this, for the size of cavities forming our CCAs we believe our system shows the highest reported Q factors for a locally tunable photonic cavity array.

We again thank the reviewer for thoroughly reading our manuscript and raising many valuable comments. We hope that we have addressed their concern in our revised manuscript.

REVIEWER COMMENTS

Reviewer #1 (Remarks to the Author):

Unfortunately, while the authors have made some additions to their work, they have not adequately addressed the most significant shortcoming - the lack of scientific impact. The present study fails to explore new physics that cannot be accessed through classical simulation. As an example, merely mentioning the potential to explore Anderson localization or the SSH model without actually doing so does not provide convincing evidence of the system's capabilities.

The current manuscript represents a small step towards the future goal of an optical quantum simulator that is easy compared to the outstanding challenges on this path. Indeed, many additional achievements need to be made before a photonic quantum simulator can become a reality. As a result, I believe that the present work is not suitable for publication in Nature Communications and would be better suited for a more specialized journal such as Physical Review A.

Reviewer #2 (Remarks to the Author):

The revised manuscript is drastically improved over the initial version. While the first draft left open many questions and seemed incomplete especially in the end, the paper now makes an effort to clearly convey the context in which the authors see their work and its applicability. While this is a positive statement, I do have to comment that this is what the initial submission should have been.

My questions were mostly answered to a satisfactory level, so from my perspective there is nothing that speaks against publication. If nature communications is a suitable platform or not is hard for me to discern. The results presented are a technological advancement and a proof of principle that the obtained control via the island heaters does work well. One has to consider if this warrants publication in a broad-interest journal or not.

Reviewer #3 (Remarks to the Author):

This paper describes a silicon photonic implementation of coupled, thermally-controlled ring resonators for the purpose of realizing tight binding Hamiltonians. Typical target applications include simulation of quantum phenomena at room temperature and using a scalable platform. The existence of an entire ecosystem of components including single photon sources and detectors at telecom wavelengths is definitely an advantage in simplifying and realizing the underlying quantum dynamics, as was already indicated in earlier references [13-15].

From the general presentation of the paper, and also from the earlier reviewer comments, it is evident that the presented results actually target several experimental challenges of realizing optical quantum simulators in a scalable platform. While the earlier reviewers have raised concerns regarding the novelty of cascaded resonator arrays, their individual electrical addressability with low thermal crosstalk is still a critical challenge to tackle, which has been demonstrated for an 8-site 1-D lattice in this paper. From this practical perspective, the presented manuscript does serve an important purpose and leads a pathway towards realistic applications which require much higher numbers of cascaded resonators. Therefore, in principle, the paper does meet the general high-impact criteria for publication in Nature Communications, in my opinion.

However, from an experimental and implementation perspective, several key details need to be

addressed. Below are my detailed comments and suggestions:

1. The spatial separation of the optical mode from the tungsten metal plays an important role in reducing optical losses in the resonators. As the authors claim that their island heaters constitutes a major part of the novelty of the presented work, some more elaborate explanations and diagrams are needed in order to clarify certain missing details.

a. Specifically, the paper should include a plot of the spatial mode distribution shown together with all the constituent waveguide materials to-scale and accurately placed. Otherwise, it is not clear what the lateral or vertical separation is between the optical mode and the tungsten heater itself. Some details can be seen in Fig S1 of the supplementary material, but the placement of tungsten with respect to the Si waveguide seems laterally inconsistent in the final few parts of that figure, which need to be fixed.

b. Once the mode distribution is plotted, the resulting propagation loss due to the placement of tungsten must be simulated and included.

2. My previous point brings up a more fundamental question regarding the placement of tungsten itself and the heater design. This is currently not discussed in the paper; and the pertaining thermal simulations are missing.

a. First of all, why have the authors chosen to place the tungsten to-the-side of the Si waveguide, instead of on-top-of the waveguide directly? This does not require a universal cladding, as the authors have said. Would it not have been possible to evaporate a thin layer of Al₂O₃ top cladding, liftoff the parts outside of the Si waveguides as usual, and place the tungsten directly over the top of the Al₂O₃ cladding, which is also then directly on top of the Si waveguide? Is that not more beneficial from a thermal efficiency perspective, since Si and W could have been placed closer together in that manner?

b. The thermal island geometry and its comparison to "cladded" version needs some clarifications:

i. Fig S3(a) and (b) should be accompanied by cross-sections of the two heaters, in order to demonstrate the differences much more clearly. It is impossible to see the important details of the heaters just from the 3D renders.

ii. The paper should include a plot of thermal distribution across the currently used waveguide cross-section, similar to the field or intensity distribution of the mode itself. This distribution should be repeated for the two cases: "cladded" and island designs.

iii. Finally, it is not obvious to me why a top Al₂O₃ layer was used, over the tungsten heater itself. The surrounding air already thermally separates the adjacent resonator, so it is not clear what the top Al₂O₃ layer of the heater sandwich achieves in this current design. Is it placed to prevent oxidation of tungsten itself?

3. With the algorithm presented in the supplementary information, one can determine the eigenvalues from the Lorentzian fits, the onsite potentials, and the hopping rates. Yet, the authors have mentioned that "The spacing between the resonators is determined by the desired hopping rate between the sites". Given the spacing between two waveguides, it is common practice in integrated photonics to calculate the coupling coefficient (and then the resulting hopping rate) between two adjacent resonators directly in simulation.

a. As the spacing is known by design, the resulting coupling should at least be calculated (numerically or analytically) in the methods section.

b. More importantly, why do the authors then need to separately extract this quantity again from their

fits? Would it not be much easier to just insert these calculated/simulated terms directly into the Hamiltonian? Are there unexpected differences between the expected coupling rate vs the experimentally measured coupling rate, potentially due to fabrication-induced changes?

4. "Since alumina is optically lossless in the telecommunication wavelength range ..." is a highly optimistic statement. The evaporation/deposition method for alumina (and many other oxides) is known to strongly influence its optical properties including both the real and imaginary parts of the refractive index [Magden et al. Optics Express 25.15 (2017): 18058-18065]. In Fig S1, it is shown that Si and Al₂O₃ share a vertical interface. The quality of this interface would also directly influence the quality factor of the resonators. Several questions arise:

a. Does the reported $Q=8.5e4$ metric take all of these factors into account? Is that the loaded quality factor of the resonators? Was that measured using a different/separate structure (resonator alone, resonator with other resonators coupled on the sides, heater included or not, etc.)?

b. In addition to just the 265nm thickness, do the authors have any details or measurements from the optical losses of the Al₂O₃ itself? How about any waveguide loss measurements of the Si/ Al₂O₃ combination?

c. While it may be sufficient for an 8-site device, does this quality factor influence the operation of much larger systems perhaps with thousands of coupled cavities? Is it possible that authors provide some asymptotic prediction on what quality factors would be necessary for operation of much larger systems and their individual eigenmode addressability, since that is what's realistically needed for more useful quantum simulations?

5. In addition to the islands, the thermal isolation is also clearly influenced by how close the heated sections of the resonators are to one another. Why have the authors then chosen not to use larger (or just longer) resonators, or which would farther separate the heaters from each other? Are there any prohibitive considerations regarding the free spectral range? Just in general, there seem to be several different ways to physically place/separate these types of heaters in various a configuration. Can the authors discuss their specific choice of geometry here?

6. From the diagram in Fig 2(a), heater n seems to be physically closer to heater $n+2$ than it is to heater $n+1$, but this is not very clear. Physical dimensions should be included on the figures to make sure these details are easily understandable. If that is the case, when heater n is switched on, would that mean the change in the onsite potential for resonator $n+2$ would be larger than that of resonator $n+1$? Can the authors comment on these details referring to the results in Fig 2(c)?

7. The expression "we establish that thermal crosstalk is already low between the nearest neighbors" needs some clarification. What is considered "low" in a target application? A change of 0.1 nm still corresponds to approximately 12.5GHz difference in frequency at the telecom band, which can still be considered very high for certain frequency-sensitive applications. What are the relevant metrics to consider in this comparison?

8. The final part of the paper demonstrates how the eigenenergies of the system can be predicted on application of a certain set of driving voltages on the heaters. For what application could this sort of procedure be necessary? Wouldn't it make sense to solve the inverse problem instead (figure out what driving voltages are necessary in order to generate a target set of eigenenergies)? Is that what would be potentially required for simulating the bosonic walks the authors mentioned in different types of lattices?

9. I agree with the earlier comments regarding the missing details on the discussion of the results, particularly regarding their prospects/usability in enabling specific applications. For instance, one of the most important aspects of programmable photonic circuits similar to those demonstrated here is

their use in dynamic applications. Therefore, some discussion of dynamic control is needed for the presented platform.

a. For instance, how about mimicking a time-dependent energy profile like one might see in a modulated material? Is the implemented system able to recreate a set of time-evolving eigenenergies?

b. Can the authors comment on the temporal characteristics of the demonstrated heaters? It is well known that this type of thermal control achieves MHz-level electrical bandwidths, unless special design considerations are applied [Atabaki et al. Optics express 18.17 (2010): 18312-18323]. Does that limit the types of quantum simulation applications the demonstrated platform could be used for?

10. In regards to programmability of eigenenergies, the authors have reported an error of 4%, in comparison to the mean hopping rate. Is it possible to implement a re-calibration procedure to further reduce this error, which can be applied after the eigenenergies are optically measured, either as a single-shot correction or in a continuous manner?

11. Also, it is true that the addressability of individual eigenmodes (or optical cavity resonances) is important for photonic implementations of quantum simulators. But does that necessarily mean that each eigenmode has to be controlled by a single knob, like an individual separate heater? Likely, for platforms like the one presented here, it would be possible to create (or learn from data) a mapping function that can reproduce a given set of eigenenergies near-perfectly, even when multiple cavity responses depend on a single heater voltage, since the underlying correlations can be easily modeled from measured data. Would that potentially reduce the fitting error, or better yet, simplify the operating procedure for lattices with larger numbers of sites?

We are very grateful to the referees for reading the manuscript and providing many insightful comments. Our point-to-point responses to the referee's comments appear below.

Reviewer #1 (Remarks to the Author):

Unfortunately, while the authors have made some additions to their work, they have not adequately addressed the most significant shortcoming - the lack of scientific impact. The present study fails to explore new physics that cannot be accessed through classical simulation. As an example, merely mentioning the potential to explore Anderson localization or the SSH model without actually doing so does not provide convincing evidence of the system's capabilities.

The current manuscript represents a small step towards the future goal of an optical quantum simulator that is easy compared to the outstanding challenges on this path. Indeed, many additional achievements need to be made before a photonic quantum simulator can become a reality. As a result, I believe that the present work is not suitable for publication in Nature Communications and would be better suited for a more specialized journal such as Physical Review A.

Response: We would like to thank the reviewer for taking the time to read our revised manuscript. However, we respectfully disagree with the reviewer regarding the impact of our work. Our work shows scalability, programmability, and measurability of photonic CCAs for the first time, which we believe is a big step forward over current photonic quantum simulators, that are traditionally not programmable or tomographically mappable. Owing to the potential that photonics has for realizing large scale quantum simulators, we believe that our work addresses some of the major challenges that have prevented photonics from realizing this aim. We believe comparing two different quantum systems at different stages of their development is a bit unfair due to the varying challenges each system faces. However, given no quantum technology presently has a clear path to reach the utility-scale computing, it is imperative that other technologies need to be explored. Our work represents a significant advancement for photonic systems, and we strongly believe this work is suitable for publication in Nature Communications.

Reviewer #2 (Remarks to the Author):

The revised manuscript is drastically improved over the initial version. While the first draft left open many questions and seemed incomplete especially in the end, the paper now makes an effort to clearly convey the context in which the authors see their work and its applicability. While this is a positive statement, I do have to comment that this is what the initial submission should have been.

My questions were mostly answered to a satisfactory level, so from my perspective there is nothing that speaks against publication. If nature communications is a suitable platform or not is hard for me to discern. The results presented are a technological advancement and a proof of principle that

the obtained control via the island heaters does work well. One has to consider if this warrants publication in a broad-interest journal or not.

Response: We would like to thank the reviewer for a positive statement regarding our work and greatly appreciate the reviewer's help in polishing it.

Despite of progress in other quantum simulation (QS) systems, photonics enjoys the following advantages:

1. Photonics can be used at much higher temperatures as photons can maintain quantum states at room temperature. This significantly eases experiments and reduces costs.
2. Despite progress, superconducting systems are restricted to a few 10s of sites, which is okay for prototypes but to simulate real materials we ideally want a lot larger number of nodes. Access to very sophisticated processes in photonic foundries can lead to a much easier path to large scale scalability.
3. Finally, availability of single photon detectors means we can directly measure multi-photon correlations like $g_2(0)$.

However, as we alluded earlier, realization of photonic analog quantum simulators depends on realization of photonic CCAs that satisfy the need of scalability, programmability, measurability, and non-linearity. Our work shows scalability, programmability, and measurability of photonic CCAs for the first time, which is a big step forward over current photonic quantum simulators, that are traditionally neither programmable nor tomographically mappable. Hence, we believe that our work is of broad interest and apt for publication in Nature Communications as it represents a significant advance for photonic systems for QS purposes.

Reviewer #3 (Remarks to the Author):

We are grateful to the referee for carefully reading the manuscript and providing many insightful comments. Our point-to-point responses to the referee's comments appear below.

This paper describes a silicon photonic implementation of coupled, thermally-controlled ring resonators for the purpose of realizing tight binding Hamiltonians. Typical target applications include simulation of quantum phenomena at room temperature and using a scalable platform. The existence of an entire ecosystem of components including single photon sources and detectors at telecom wavelengths is definitely an advantage in simplifying and realizing the underlying quantum dynamics, as was already indicated in earlier references [13-15].

From the general presentation of the paper, and also from the earlier reviewer comments, it is evident that the presented results actually target several experimental challenges of realizing optical quantum simulators in a scalable platform. While the earlier reviewers have raised concerns regarding the novelty of cascaded resonator arrays, their individual electrical addressability with low thermal crosstalk is still a critical challenge to tackle, which has been

demonstrated for an 8-site 1-D lattice in this paper. From this practical perspective, the presented manuscript does serve an important purpose and leads a pathway towards realistic applications which require much higher numbers of cascaded resonators. Therefore, in principle, the paper does meet the general high-impact criteria for publication in Nature Communications, in my opinion.

Response: We thank the reviewer for the positive comment towards our work for achieving an 8-site tight binding photonic simulator and for appreciating its significance to a broad scientific audience. We sincerely appreciate the detailed review and have responded to all the criticisms. We believe these modifications made the paper stronger and more accessible to a broad audience.

However, from an experimental and implementation perspective, several key details need to be addressed. Below are my detailed comments and suggestions:

1. The spatial separation of the optical mode from the tungsten metal plays an important role in reducing optical losses in the resonators. As the authors claim that their island heaters constitutes a major part of the novelty of the presented work, some more elaborate explanations and diagrams are needed in order to clarify certain missing details.

a. Specifically, the paper should include a plot of the spatial mode distribution shown together with all the constituent waveguide materials to-scale and accurately placed. Otherwise, it is not clear what the lateral or vertical separation is between the optical mode and the tungsten heater itself. Some details can be seen in Fig S1 of the supplementary material, but the placement of tungsten with respect to the Si waveguide seems laterally inconsistent in the final few parts of that figure, which need to be fixed.

b. Once the mode distribution is plotted, the resulting propagation loss due to the placement of tungsten must be simulated and included.

Response: We thank the reviewer for the suggestion. We have added an additional section to the Supplementary Information titled: S1. Optical design & considerations. The first subsection S1A now includes the plotted mode profile to scale for both the cases with an overlay of the lateral geometry and the respective losses. We have also adjusted the fabrication flow figure. The updated fabrication flow and first subsection of Optical design & considerations in SI is as follows:

Fabrication:

Fig. S1. Fabrication flow. The CCA was defined on a silicon on insulator (SOI) chip via electron-beam lithography using *HSQ* resist. Cl_2 gas was then used to etch the structures. The resist was then removed using dilute *BOE*. For the deposition steps positive tone *PMMA* resist was used for patterning. Bottom Al_2O_3 layer is $265nm$ thick and the sputtered tungsten (*W*) layer making up the heating elements is $150nm$ thick. The contact pads are made up of $25nm$ *Ti* / $325nm$ *Pt* layers. The final Al_2O_3 cladding over the islands is $300nm$ thick.

SIA: Waveguide Modes:

Our device consists of racetrack resonators fabricated on 220 nm thick silicon on insulator platform. The resonators have two differing waveguide sections: (i) a standard ridge waveguide section which is 554 nm wide with an 82° slant and (ii) and an embedded waveguide section where resonator makes thermal contact with the heater through the bottom alumina layer of the island heaters. The simulated optical mode profile (using ANSYS Lumerical MODE) for both these sections is plotted in Fig. S2. The simulated waveguide losses are $9.2 \times 10^{-13} dB/cm$ and $7.8 \times 10^{-9} dB/cm$ respectively. As evident the loss is negligible even for the heater integrated design as we were able to utilize the bottom alumina platform (which is 265 nm thick) to place the tungsten heater both above the plane of the optical mode and at a far enough ($1.5\ \mu m$) lateral distance to ensure minimal increase in the absorption losses.

Fig. S2. Waveguide mode profiles ($|E|^2/|E|_{max}^2$). (a) Profile of the waveguide mode in the sections without any cladding layers. The effective index of the mode is $n_{eff} = 2.556$ and the group index is $n_g = 4.13$. The simulated loss is $9.213 \times 10^{-13} \text{ dB/cm}$. A lateral undercut of $\sim 27 \text{ nm}$ occurs during the BOE strip of HSQ resist. (b) Profile of the waveguide mode where resonator makes thermal contact with the island heaters with the tungsten heating element being $1 \mu\text{m}$ wide. The effective index of the mode is $n_{eff} = 2.61$ and the group index is $n_g = 4.006$. The simulated loss is $7.75 \times 10^{-9} \text{ dB/cm}$.

We have further added a line to the text referencing this section:

Experimentally, we implement a Hamiltonian with 8 nodes via a CCA made up of 8 strongly coupled racetrack resonators fabricated on a silicon-on-insulator platform using 220 nm silicon on top of $3 \mu\text{m}$ thick silicon oxide (Fig. 1a). The spacing between the resonators is determined by the desired hopping rate between the sites for the tight binding Hamiltonians being implemented (see supplementary information for optical mode profiles and hopping rate calculation).

2. My previous point brings up a more fundamental question regarding the placement of tungsten itself and the heater design. This is currently not discussed in the paper; and the pertaining thermal simulations are missing.

a. First of all, why have the authors chosen to place the tungsten to-the-side of the Si waveguide, instead of on-top-of the waveguide directly? This does not require a universal cladding, as the authors have said. Would it not have been possible to evaporate a thin layer of Al_2O_3 top cladding, liftoff the parts outside of the Si waveguides as usual, and place the tungsten directly over the top of the Al_2O_3 cladding, which is also then directly on top of the Si waveguide? Is that not more beneficial from a thermal efficiency perspective, since Si and W could have been placed closer together in that manner?

Response: In our design the distance between the Si waveguide and the tungsten heater is $1.5 \mu\text{m}$. This is kept to be large enough to ensure that the introduction of the metal heater does not introduce any additional absorptive losses. As the reviewer mentions, fabrication wise it is entirely possible to evaporate a layer of Al_2O_3 bottom cladding, liftoff the parts outside of the Si waveguide and place the tungsten directly over the top of the Al_2O_3 cladding, which is also then directly on top of the Si waveguide. However there are several subtle problems here: if we are to keep the physical spacing between the heater and the Si waveguide same as before ($1.5 \mu\text{m}$) to ensure similar absorptive losses then placing on top will have only have a slight advantage in steady state operation but will come at cost of a less reliable fabrication process involving lifting off a thick $1.5 \mu\text{m}$ Al_2O_3 layer which would be plagued by several complications including but not limited to lithography of very thick resists and less reliable alignment. Additionally, if the island was made up of $1.5 \mu\text{m}$ thick alumina layer, even though the chip would be cladding free, for future material integration purposes the chip design would be a lot less suitable to integrate with excitonic materials due to sharp topography of the chip area. But, if we were to place the heater closer to the

waveguide, than the suggested design will indeed be a lot more thermally efficient but will come at a cost of a significantly higher loss. Our bottom layer of alumina was designed to be slightly thicker than Si layer, and this ensured that the heater is out of the plane of the optical mode and laterally far away so as to cause minimal addition of losses.

b. The thermal island geometry and its comparison to “cladded” version needs some clarifications:

i. Fig S3(a) and (b) should be accompanied by cross-sections of the two heaters, in order to demonstrate the differences much more clearly. It is impossible to see the important details of the heaters just from the 3D renders.

ii. The paper should include a plot of thermal distribution across the currently used waveguide cross-section, similar to the field or intensity distribution of the mode itself. This distribution should be repeated for the two cases: “cladded” and island designs.

Response: We have updated the thermal simulation section of the SI to have a better 3D model and added a figure demonstrating the lateral cross section of heater overlaid on the thermal distribution for both the designs at a specific voltage. For ease of reviewer the edited section is as follows:

Section S3: Thermal crosstalk simulations

For a racetrack resonator we know that $n_{eff}l = m\mu_n^\lambda$, where n_{eff} is the refractive index of the racetrack resonator, l is the length of the resonator, μ_n^λ is the onsite potential (in wavelength units) and $m \in Z$. Let a segment of length x be affected by change in temperature, such that its refractive index is given by $n(x)$. Resonance condition for the resonator then becomes:

$$n_{eff}(l - x) + \int n(x)dx = m(\mu_n^\lambda + \Delta\mu_n^\lambda) \quad (S36)$$

Removing the constant terms gives

$$\int (n(x) - n_{eff})dx = \int \Delta(n(x))dx = m(\Delta\mu_n^\lambda) \quad (S37)$$

Assuming a constant thermo-optic coefficient dn/dT we can write $\Delta n(x) = \rho\Delta T(x)$. Consequently,

$$\int \rho\Delta T(x)dx = m(\Delta\mu_n^\lambda) \quad (S38)$$

This implies shift in onsite potential $\Delta\mu_n^\lambda \propto \int \Delta T(x)dx$.

Next, to estimate this effect of thermal crosstalk in our system and compare it with typically used thermo-optic (TO) heaters we perform a set of thermal simulations using ANSYS Lumerical HEAT (Fig. S6). In Fig. S6(a) we have the schematic depicting a conventional TO heater, where the metallic heating element sits directly on top of the resonator segment separated by a uniform and universal $1.5\mu m SiO_2$ cladding. In Fig. S6(c) we have the island TO heater design we used for our device. For both these cases we have the CCA made up of only 3 sites with the heater

placed on the middle site (n^{th}). We then record the temperature profile in the shorter straight segment (highlighted in yellow) of the racetrack resonators for the n^{th} and $(n + 1)^{th}$ sites as we vary the voltage applied V_n across the heater from $0V$ to $0.78V$.

Fig. S6. Thermal simulation results. (a) Schematic depicting the device with conventional TO heater sitting directly over the resonator separated by $1.5 \mu\text{m}$ thick oxide cladding (label: Cladded). (b) Temperature profile (in K) across the cladded waveguide (cross section denoted by red dashed line in the schematic) when $V_n = 0.56V$. (c) Schematic depicting a device equipped with the island TO heater used in our paper (label: Island). (d) Temperature profile (in K) across the island waveguide (cross section denoted by red dashed line in the schematic) when $V_n = 0.56V$. (e) Plot comparing average temperature across the straight segment (in yellow) of the $(n + 1)^{th}$ resonator. (f) Plot comparing average temperature across the straight segment (in yellow) of the n^{th} resonator. (g) Plot comparing η factor in % for the two device designs. (purple: cladded, pink: island).

Then in Fig. S6(b), (d) we plot temperature profile (in K) across the waveguide cross section (denoted by the red dashed line in the corresponding schematic) for both the designs when 0.56 V is applied across the heater. In Fig. S6(e) we plot the average temperature across the segments given by

$$Avg(T) = \frac{\int T(x)dx}{\int dx} \quad (S39)$$

for neighboring $(n + 1)^{th}$ site. Similarly, in Fig. S6(f) we plot the average temperature across the segment for n^{th} site. It is clear from these plots that even though both heater designs cause a similar increase in the onsite temperature (n^{th}), the average temperature in the neighboring sites diverges rapidly between the designs with the temperature difference already greater than 24K at 0.78V. To study the difference in effects of this thermal crosstalk we then define a dimensionless parameter η given by

$$\eta = \frac{\Delta\mu_{n+1}^\lambda}{\Delta\mu_n^\lambda} \quad (S40)$$

Using Eq. S38 we can write the above as

$$\eta = \frac{\Delta\mu_{n+1}^\lambda}{\Delta\mu_n^\lambda} = \frac{\int \Delta T_{n+1}(x)dx}{\int \Delta T_n(x)dx} = \frac{\int (T_{n+1}(x) - 300)dx}{\int (T_n(x) - 300)dx} \quad (S41)$$

In Fig. S6(g) we plot the η (as %) for both the designs. We can see from the plot that $\eta_{cladded} \sim 0.037$ and $\eta_{island} \sim 0.018$. This implies that the islands TO heaters outperform the typical TO heaters in reducing the effects of thermal crosstalk in the device by $\sim 50\%$.

In general, we do not expect the η values to vary with the voltage V_n (see Section S4) as also verified by the plots.

iii. Finally, it is not obvious to me why a top Al2O3 layer was used, over the tungsten heater itself. The surrounding air already thermally separates the adjacent resonator, so it is not clear what the top Al2O3 layer of the heater sandwich achieves in this current design. Is it placed to prevent oxidization of tungsten itself?

Response: The reviewer is indeed correct; the top Alumina cladding is added to protect the tungsten heater. We have added the following line to the paper:

Additionally, the top alumina layer acts as a protective layer against oxidation for the tungsten heating element²⁰.

3. With the algorithm presented in the supplementary information, one can determine the eigenvalues from the Lorentzian fits, the onsite potentials, and the hopping rates. Yet, the authors have mentioned that “The spacing between the resonators is determined by the desired hopping rate between the sites”. Given the spacing between two waveguides, it is common practice in integrated photonics to calculate the coupling coefficient (and then the resulting hopping rate) between two adjacent resonators directly in simulation.

a. As the spacing is known by design, the resulting coupling should at least be calculated (numerically or analytically) in the methods section.

Response: We have added subsection S1B to the SI showing the procedure to design hopping rates and the interplay between the physical spacing between the resonators and their coupling strength. The added subsection is included after response of the subpart b. of the question.

b. More importantly, why do the authors then need to separately extract this quantity again from their fits? Would it not be much easier to just insert these calculated/simulated terms directly into the Hamiltonian? Are there unexpected differences between the expected coupling rate vs the experimentally measured coupling rate, potentially due to fabrication-induced changes?

Response: The reviewer is indeed correct. Both the onsite potential and the hopping rates between the resonators are very susceptible to minor variations induced due to fabrication processes. Hence if we are implementing a Hamiltonian to study a particular physical effect, we need to ensure that the realized Hamiltonian is close enough so that the obtained results are accurate. Additionally, efficacy of all control protocols is also strongly dependent on the accuracy of the initially determined Hamiltonian matrix. In context of simulating Hamiltonians this problem is not new and in microwave superconducting circuits simulator community, researchers have come up with their own tomography protocols relevant to the platform (see Hangleiter, Dominik, et al. arXiv:2108.08319 (2021); Ma, Ruichao, et al. Physical Review A 95.6 (2017): 062120). In our work we develop a tomography algorithm that is suitable for photonic CCAs which have relatively higher losses and have restricted access to only sites at the outermost boundaries. We have addressed this point in the last paragraph of the newly added subsection ‘S1B: Hopping rates’ in the SI. The subsection is added below for ease of the reviewer:

S1B: Hopping rates:

Another aspect of the optical design deals with the interplay of the physical spacing between the resonators and the corresponding hopping rates between the sites. As the waveguides forming the coupling region of neighboring resonators in a CCA lie in close proximity, their respective modes overlap with each other to create two coupled modes as depicted in Fig. S3. The coupling constant k can be estimated numerically by simulating the eigenmodes of the two adjacent waveguides and is given by the following relation:

$$k = \sin\left(\frac{\pi\Delta n_{eff}}{\mu\lambda}L_{coupling}\right) \quad (S1)$$

where Δn_{eff} is the difference between the effective indices of the coupled waveguide modes, $L_{coupling}$ is the coupling length of the resonators and $\mu\lambda$ is the resonant wavelength of the resonator. Given the coupling constant between the coupled waveguide sections, the corresponding

hopping rate between the resonators (assuming no disorder) can then be calculated using the following relation^{1,2} :

$$J = \frac{kv_g}{L_{resonator}} \quad (S2)$$

where v_g denotes the group velocity of the waveguide mode and the $L_{resonator}$ denotes the length of resonator. Combing these two equations, the estimated hopping rate between the resonators can be written as

$$J = \sin\left(\frac{\pi\Delta n_{eff}}{\mu\lambda}L_{coupling}\right)\frac{c}{n_g L_{resonator}} \quad (S3)$$

Note that we have substituted $v_g = c/n_g$ where n_g denotes the group index of the waveguide mode and the hopping rate J is obtained in units of rad/s .

Substituting for the values we get $J \approx 69.1 GHz$. From our tomography algorithm the experimentally measured hopping rates between the resonators in the CCA were found to be 72.1 GHz, 75.7 GHz, 75.1 GHz, 77.9 GHz, 69.5 GHz, 70 GHz, 64.5 GHz giving a mean hopping rate of $J_{mean} \approx 72 GHz$ with a standard deviation of $\sigma_j = 4.2 GHz$. Additionally, on the same chip we fabricated two photonic molecules ($N = 2$) and the reconstructed hopping rate for these after measurement and running the tomography algorithm was found to be 68.2 GHz and 68.8 GHz respectively. We note that even though the estimated hopping rate is in close agreement of the measured values, relation S3 is only strictly true in absence of onsite disorder. Additionally, the hopping rates are very sensitive to minor changes in difference of refractive indices of the coupled modes (for small k , $J \propto \Delta n_{eff}$) which can arise due to a variety of fabrication induced local variations. As nanofabrication processes are always susceptible to inherent disorder, the above strongly emphasizes the need of using the tomography algorithm to accurately reconstruct the actual hopping rates (and onsite potentials) of the device post-fabrication.

Fig. S3. Coupled waveguide mode profiles ($|E|^2/|E|^2_{max}$). (a) Mode profile of the first coupled waveguide mode $n_{eff} = 2.568$. (b) Mode profile of the second coupled waveguide mode, $n_{eff} = 2.5525$. $\Delta n_{eff} \approx 0.0155$.

We have further added a line to the text referencing this section:

Experimentally, we implement a Hamiltonian with 8 nodes via a CCA made up of 8 strongly coupled racetrack resonators fabricated on a silicon-on-insulator platform using 220 nm silicon on top of 3 μm thick silicon oxide (Fig. 1a). The spacing between the resonators is determined by the desired hopping rate between the sites for the tight binding Hamiltonians being implemented (see supplementary information for optical mode profiles and hopping rate calculations).

4. *“Since alumina is optically lossless in the telecommunication wavelength range ...” is a highly optimistic statement. The evaporation/deposition method for alumina (and many other oxides) is known to strongly influence its optical properties including both the real and imaginary parts of the refractive index [Magden et al. Optics Express 25.15 (2017): 18058-18065]. In Fig S1, it is shown that Si and Al2O3 share a vertical interface. The quality of this interface would also directly influence the quality factor of the resonators.*

Response: We have updated the line in paper as follows:

Since alumina is typically optically lossless in the telecommunication wavelength range (see supplementary information for ellipsometry data), the islands also allow for placing the tungsten heaters at an adequate distance from the racetrack resonators.

Several questions arise:

a. *Does the reported $Q=8.5e4$ metric take all of these factors into account? Is that the loaded quality factor of the resonators? Was that measured using a different/separate structure (resonator alone, resonator with other resonators coupled on the sides, heater included or not, etc.)?*

Response: Because we are doing a full tomography, we can accurately determine the loss rates in Hamiltonian for each site. The reported Q factor in the manuscript is the highest attained one of ~ 71,000 (we have fixed this calculation error in the manuscript). The Q factors for individual cavities include effect of all possible losses including scattering losses due to mode mismatch, absorptive losses due to heaters, bending losses but does not coupling rates to gratings (as this a control parameter)), whereas the Q factors of eigenmodes includes all the above losses as well as effect of coupling to the grating couplers. We have added a subsection ‘S2A Quality factor extraction & comparison’ to the SI detailing how the intrinsic Q factor is extracted, the average Q factor obtained and Q factors of the supermodes are calculated. The first part of the subsection is as follows:

S2A: Quality factor extraction & considerations:

The relevant Q-factors for us here are the Q-factors of the eigenmodes of the system which effect Hamiltonian dynamics and the number of modes we can address for a given hopping rate, and the

intrinsic Qs of the cavities which act as a barometer of loss in our system due to integration with island heaters. These can be calculated as follows: In the step (i) of the tomography algorithm we fit each of the supermodes to a complex Lorentzian and hence obtain both their eigenfrequencies and linewidths. The following table shows the quality factor of each of the eigenmodes calculated using the relation:

$$Q_\alpha = \frac{\omega_\alpha}{2\beta_\alpha} \quad (\text{S32})$$

where Q_α denotes the Q-factor of the eigenmode α .

Table. S1. Q-factors of eigenmodes. Quality factors of the 8 supermodes of the CCA. The average linewidth of the eigenmodes is 7.41GHz and the average Q-factor is $\text{Avg}(Q_\alpha) \sim 2.8 \times 10^4$.

Q_0	Q_1	Q_2	Q_3	Q_4	Q_5	Q_6	Q_7
32432	32380	28314	21609	20394	23738	31814	31746

Additionally, after going through the entire tomography algorithm, we know all the complex onsite potentials lying along the diagonal of the reconstructed Hamiltonian H_{eff} . Hence, estimating the intrinsic the Q-factors of the cavities not directly coupled to the grating couplers ($n \neq 1, N - 1$) is straightforward as their complex potentials can be expressed as ($\tilde{\mu}_n = \mu_n - j\frac{\kappa_n}{2}$, $n \neq 0, N - 1$). The intrinsic Q-factor can then be calculated as:

$$Q_{intrinsic} = \frac{\text{Re}(\tilde{\mu}_n)}{-2\text{Im}(\tilde{\mu}_n)} \quad \forall n \neq 1, N - 1 \quad (\text{S33})$$

The following table then shows the intrinsic Qs for the realized CCA:

Table. S2. Q-factors of individual cavities. Intrinsic quality factors of the 6 internal cavities ($n \in [1,6]$).

Q_1	Q_2	Q_3	Q_4	Q_5	Q_6
49821	28467	32146	32148	31137	71166

Finally, as we also obtained the coupling rate to the grating coupler from the algorithm ($\gamma \approx 11.6\text{GHz}$), we can use it to estimate the average loss rate in the CCA from the following relation:

$$\text{Avg}(\kappa_n) = 2 \left(- \sum_{n=0}^{N-1} \text{Im}(\tilde{\mu}_n) - \gamma \right) \quad (\text{S34})$$

Then the typical intrinsic Q can be estimated as

$$Expected(Q_{intrinsic}) = \frac{Avg(Re(\tilde{\mu}_n))}{Avg(\kappa_n)} \quad (S35)$$

which for our device gives $Avg(\kappa_n) = 4.49 \text{ GHz}$, $Expected(Q_{intrinsic}) \approx 4.43 \times 10^4$. We also fabricated some add-drop filters (single racetrack resonators coupled to grating couplers on either side) on the same chip and measured their Q-factors before integration of the island heaters. The measured data is presented in the table below:

Table. S3. Q-factors of single racetrack resonators (in add-drop configuration) before the addition of island heaters. Q_L denotes the loaded Q-factor, $Q_{intrinsic}$ denotes the unloaded Q-factor, BW denotes the total bandwidth = loss rate + $2 \times$ coupling rate.

No.	Q_L	$Q_{intrinsic}$	$BW \text{ (GHz)}$
1	8324	99920	23.96
2	8701	257775	22.9
3	8949	192177	22.28

The Q-factor of the add-drop filters post integration of the island heaters was found to be similar to the Q-factor reported above for the main device. For example, for the add-drop filter *No. 3* the Q-factors post integration were: $Q_L = 7073$, $Q_{intrinsic} = 45067$ and $BW = 28.18 \text{ GHz}$. The intrinsic Q-factor matches very well with the $Expected(Q_{intrinsic})$ calculated from the CCA device.

We have also updated the line mentioned in the paper to make it clearer:

In this work, we tackle these problems by engineering a silicon photonic CCA made of high-Q (intrinsic Q s up to $\sim 7.1 \times 10^4$) racetrack resonators with thermally controllable onsite potential using specially designed thermo-optic (TO) island heaters.

b. In addition to just the 265nm thickness, do the authors have any details or measurements from the optical losses of the Al2O3 itself? How about any waveguide loss measurements of the Si/Al2O3 combination?

Response: We used ellipsometry to measure the complex refractive index of our evaporated alumina. The imaginary part of the refractive index was found to be 0. We have included the ellipsometry results now in subsection S2A of the SI as:

Considerations:

Fig. S5. Ellipsometry data of evaporated alumina. The refractive index of evaporated alumina is measured using ellipsometry. The imaginary part of the index is negligible, denoting that alumina is lossless in the window of operation of our CCA. $n(1505 \text{ nm}) = 1.57532$.

We have also updated a line in the paper referencing this section:

Since alumina is typically optically lossless in the telecommunication wavelength range (see supplementary information for ellipsometry data), the islands also allow for placing the tungsten heaters at an adequate distance from the racetrack resonators.

c. While it may be sufficient for an 8-site device, does this quality factor influence the operation of much larger systems perhaps with thousands of coupled cavities? Is it possible that authors provide some asymptotic prediction on what quality factors would be necessary for operation of much larger systems and their individual eigenmode addressability, since that is what's realistically needed for more useful quantum simulations?

Response: The quality factor does indeed influence the operation of much larger systems particularly when it comes to the tomography algorithm and individual addressability to eigenmodes. We know that the energy band of a 1D infinite tight binding model is confined within $+2J$ to $-2J$ about the central potential. As individual addressability to modes is desired, the maximum number of cavities that can be present given the average linewidth of the resonator can be ball parked as $N_{cav} = \frac{4J}{Avg(\kappa_n)}$ where we assume small coupling rates to the input/output ports.

We can invert this equation to get an idea about the quality factors desired for say 1000 cavities.

$$Q_{desired} \sim N_{cav} \frac{\mu}{4J} = 1000 \frac{199.29}{4 \times 0.072} \approx 6.9 \times 10^5$$

While this is an order of magnitude higher than the current reported Q-factor, exploiting further sophisticated techniques in optical design like using Euler bends in racetracks and adding a mode matching transition between the cladded and uncladded areas as well as using better fabrication

techniques are amongst some of the potential pathways which can be used to approach this number in future works.

We have added the following to the newly added subsection S2A of the SI:

Considerations:

...

From our simulations in the above sections and measured data, we expect the major source of loss in our device to be the scattering loss due to mode mismatch between the cladded and uncladded sections of the racetrack resonator. As in subsection S1A, we used mode simulations to calculate the worst-case power coupling between the modes (by estimating their overlap) to be 0.99675; which is on the lower side, hence confirming our hypothesis. Additionally, based on our analysis we can perform a back of the hand calculation to estimate the kind of Q-factors we would ultimately desire in order to realize very large-scale systems made up of say 1000 cavities. We know that the energy band of a 1D infinite tight binding model is confined within $+2J$ to $-2J$ about the central potential³. As individual addressability to eigenmodes is desired, the maximum number of cavities that can be present given the average linewidth of the resonator can then be ballparked as $N_{cav} = \frac{4J}{Avg(\kappa_n)}$ where we assume small coupling rates to the input/output ports. We can then invert this equation to get an idea about the Q-factors desired for a CCA consisting of 1000 cavities:

$$Q_{desired} \sim N_{cav} \frac{\mu}{4J} = 1000 \frac{199.29}{4 \times 0.072} \approx 6.9 \times 10^5$$

While this is an order of magnitude higher than the current reported Q-factor, in future works we aim to reduce the loss in the constituent cavities by designing better mode matched waveguide sections. We also plan to incorporate Euler bends⁴ in our racetrack resonators and further optimize the fabrication processes to improve the Q-factors of our devices.

5. In addition to the islands, the thermal isolation is also clearly influenced by how close the heated sections of the resonators are to one another. Why have the authors then chosen not to use larger (or just longer) resonators, or which would farther separate the heaters from each other? Are there any prohibitive considerations regarding the free spectral range? Just in general, there seem to be several different ways to physically place/separate these types of heaters in various a configuration. Can the authors discuss their specific choice of geometry here?

Response: We mention in the paper that “the small device footprint necessary to obtain small mode volumes for each cavity and ensure strong coupling between the cavities¹⁶”. The reason we want to have the optical mode be confined to a small mode volume is to ensure that our design is suitable

for future access to optical nonlinearity (see D. Lukin et al. Physical Review X 13.1 (2023): 011005.) and allows us to have sufficient coupling between sites to offset the effect of disorder while maintaining large free spectral ranges. Though longer resonators will have higher Qs and can allow us to place heaters far away from each other, they also suffer from huge mode volumes (and hence much a lower potential for nonlinear interactions), lower hopping rates and smaller free spectral ranges. We have edited the line and added a reference to make it clearer as:

“the small device footprint necessary to obtain small mode volumes for each cavity and ensure strong coupling between the cavities while maintaining large free spectral ranges^{16,19}”

We have also edited a line in the discussion to further highlight this as:

While we did not demonstrate any non-linearity, our CCA with its small footprint, high Q-factors and a cladding free design has the potential to enable integration with excitonic materials³⁰, defect centers^{31,32} and possibly reach single photon non-linear regime³³⁻³⁵.

6. From the diagram in Fig 2(a), heater n seems to be physically closer to heater $n+2$ than it is to heater $n+1$, but this is not very clear. Physical dimensions should be included on the figures to make sure these details are easily understandable. If that is the case, when heater n is switched on, would that mean the change in the onsite potential for resonator $n+2$ would be larger than that of resonator $n+1$? Can the authors comment on these details referring to the results in Fig 2(c)?

Response: Even though heater at n is closer to the heater at $n + 2$ than $n + 1$, the relevant distance for Fig. 2(c) is the distance of heater h_n to the racetrack resonators at site $n, n + 1, n + 2$. This distance follows the relation: distance to $n <$ distance to $n + 1 <$ distance to $n + 2$. The effect of thermal crosstalk when only one heater is turned ON is because of the heater temperature on the racetrack resonators and not the temperatures of other heaters which are turned OFF at this point. If, however other heaters are turned ON we would indeed need to model this effect, which is precisely what we do through the γ coefficients in Fig 3. Then though as these are correlation matrix terms, relative values of γ s are much harder to predict and hence we used numerical modelling on top of the measured data to fit for these.

We have updated the figure to convey the distance between heaters and resonators. Attached here for ease of the reviewer:

Fig. 2 Electrical Characterization. (a) Device schematic depicting the electrical characterization as voltage V_n is applied to the n^{th} site while measuring the transmission spectrum ($d_x = 14.66 \mu\text{m}$). (b) Exploded view of the TO island heaters. The heater consists of a tungsten element sandwiched between alumina layers. Inset shows a false colored SEM image (scale bar: $2 \mu\text{m}$) of a typical TO island (yellow: tungsten, pink: alumina, teal: silicon). (c) Plot showing the effect of heaters $[h_n]$ on the potential profile across the device. The x-axis denotes the heater index h_n switched ON for a particular set of measurements and the y-axis represents the change in potential profile $[\Delta\mu_n^\lambda]$. The voltage applied for the measurement (V_n) across heater h_n is mapped to the color of the circular surface and the corresponding change in potential is denoted by the radii of the circle encompassing the surface (0.25 nm of change is depicted by radii of the circle in the scale bar).

We have also added some relevant dimensions in the description of Fig 1 as

Fig. 1 Hamiltonian Tomography. (a) Optical image of the electrically controlled CCA depicting the wiring structure, optical micrograph of the CCA (scale bar: $10 \mu\text{m}$). The constituent racetrack resonators are characterized by longer straight segments that are $12 \mu\text{m}$ long, shorter straight segments that are $4 \mu\text{m}$ long, and a bending radius of $5 \mu\text{m}$.

7. The expression “we establish that thermal crosstalk is already low between the nearest neighbors” needs some clarification. What is considered “low” in a target application? A change of 0.1 nm still corresponds to approximately 12.5 GHz difference in frequency at the telecom band, which can still be considered very high for certain frequency-sensitive applications. What are the relevant metrics to consider in this comparison?

Response: The relevant metric here is the error in comparison to the hopping rate which is the relevant physical rate of the tight-binding Hamiltonian. We can see the errors in the predicted eigenvalues as a proxy to unwanted disorders in the onsite potentials with respect to the hopping rate. We know that if the ratio: *disorder/hopping rate* $\ll 1$ we will have the realized Hamiltonian behave faithfully to what we expect. (see, ACS Photonics 9.2 (2022): 682-687.)

8. The final part of the paper demonstrates how the eigenenergies of the system can be predicted on application of a certain set of driving voltages on the heaters. For what application could this sort of procedure be necessary? Wouldn't it make sense to solve the inverse problem instead (figure out what driving voltages are necessary in order to generate a target set of eigenenergies)? Is that what would be potentially required for simulating the bosonic walks the authors mentioned in different types of lattices?

Response: The control model discussed in the paper is used to demonstrate the controllability of the CCA. We wanted to highlight that the tomography algorithm used to reconstruct the initial Hamiltonian was accurate and the effects of the thermal crosstalk were low enough; that we could predict the eigenvalues of the modified Hamiltonian with a forward control algorithm based on a

limited number of correlations terms to a remarkable accuracy. It was used to demonstrate the potential of the device to be used later when tailored to specific needs to realize a range of Hamiltonians. As the reviewer correctly surmises that the inverse problem to this is indeed also of great interest but it would not have demonstrated the benefits of our approach, as it would involve fitting for inverse correlations in data to get voltage values to obtain specific eigenvalues which is a far broader vector space and might not have unique solutions as two different Hamiltonians can have same eigenvalues (similar matrices). Additionally, a brute force solution to the inverse problem will in its inner layers try to fit for the initial Hamiltonian and require far more data to be trained on. Ideally, what we would want is, that once we have intuition of the forward function (as demonstrated in the paper), we would like to analytically formulate the form of the inverse function and use our knowledge of the initial Hamiltonian to fit for this inverse. While this was out of scope of this current work, we are currently working on it for our future work in this area.

9. I agree with the earlier comments regarding the missing details on the discussion of the results, particularly regarding their prospects/usability in enabling specific applications. For instance, one of the most important aspects of programmable photonic circuits similar to those demonstrated here is their use in dynamic applications. Therefore, some discussion of dynamic control is needed for the presented platform.

a. For instance, how about mimicking a time-dependent energy profile like one might see in a modulated material? Is the implemented system able to recreate a set of time-evolving eigenenergies?

b. Can the authors comment on the temporal characteristics of the demonstrated heaters? It is well known that this type of thermal control achieves MHz-level electrical bandwidths, unless special design considerations are applied [Atabaki et al. Optics express 18.17 (2010): 18312-18323]. Does that limit the types of quantum simulation applications the demonstrated platform could be used for?

Response: The reviewer is correct in stating that our type of thermal control achieves MHz-level electrical bandwidth. Typically, there exists a large class of tight-binding Hamiltonians of interest which can be simulated without requiring dynamical control over the parameters like the Hofstadter lattice Hamiltonians (M. Hafezi et al. Nature Photonics 7, 1001 (2013), P. Roushan et al. Science 358.6367 (2017).), strained graphene Hamiltonians (A. Youssefi et al. Nature 612, 666 (2022)), quantized quadrupole phase Hamiltonian (S. Mittal et al. Nature Photonics 13, 692 (2019).) and SSH Hamiltonians (ACS Photonics 9.2 (2022): 682-687). That said there also exists a class of Floquet type Hamiltonians which require time modulated parameters to study certain effects like the Haldane Quantum Hall effect (M. Minkov and V. Savona. Optica 3.2 (2016): 200-206). This Hamiltonian for example requires modulation rates of the onsite potentials on the order of the hopping rates present in the system (few GHzs). Like above, typically, time modulated Hamiltonians that are of interest require the modulation rate of the parameters to be comparable to other relevant rates (hopping rates, loss rates, on site potentials) which is not possible to do in

optics with MHz level TO modulation. Conventional effects used in photonics for dynamic applications like the electro-optic modulation/acousto-optic modulation while can reach these GHz level speeds, are typically restricted to smaller amount of DC modulation and are much harder to scale due to experimental challenges (Zhang, Mian, Nature Photonics 13.1 (2019): 36-40.). Additionally, these faster devices at times will themselves utilize a TO tuning element for frequency alignments (Hu, Yaowen, et al. 599.7886 (2021): 587-593.) as the TO elements are ideal for realizing static potential profiles. We have added to the last paragraph of the discussion to include the above as

Our device can already be used to simulate a number of single-particle physical effects like Anderson localization¹³ and the Su–Schrieffer–Heeger (SSH) model²⁴. One potential disadvantage of using TO heaters is that our dynamic modulation rates are limited to the MHz regime²¹ which rules out the possibility to implement models like the Haldane quantum Hall effect²⁵ requiring modulation of onsite potentials at rates comparable to the mean hopping rate ($\sim GHz$). However, what TO heaters might lack, they make up for it by allowing a larger range of static modulation and ease of scalability in comparison to say electro-optical modulators which might be much faster but present far more challenges when it comes to scaling to a larger number of sites²⁶. Looking ahead, our TO island heaters equipped CCAs are hence perfectly suited for implementing a large class of Hamiltonians that do not require very high-speed modulation, such as the Hofstadter Hamiltonian^{3,27}, SSH Hamiltonian^{16,24} and non-Hermitian topological Hamiltonians²⁸, among others. Further, leveraging the immense scaling potential of photonics, operating in a linear regime the current CCA can be scaled to sizes where it can be used to study classical and quantum bosonic walks and solid-state lattice band structures²⁹.

10. In regards to programmability of eigenenergies, the authors have reported an error of 4%, in comparison to the mean hopping rate. Is it possible to implement a re-calibration procedure to further reduce this error, which can be applied after the eigenenergies are optically measured, either as a single-shot correction or in a continuous manner?

Response: Control algorithms involving feedback and data-learning based control are indeed the natural next step for our project and we are currently looking into these for our future works. However, these were out of the scope of our current project, where we wanted to demonstrate CCAs integrated with thermal heaters which have thermal crosstalk low enough that they allow forward control of the resulting Hamiltonian.

We have added a line in the paper to allude to the above discussion:

Additionally, adopting more complex control algorithms involving feedback control^{36,37} and data-based learning³⁸ in future works can help to improve the accuracy of the realized Hamiltonians further.

11. Also, it is true that the addressability of individual eigenmodes (or optical cavity resonances) is important for photonic implementations of quantum simulators. But does that necessarily mean that each eigenmode has to be controlled by a single knob, like an individual separate heater? Likely, for platforms like the one presented here, it would be possible to create (or learn from data) a mapping function that can reproduce a given set of eigenenergies near-perfectly, even when multiple cavity responses depend on a single heater voltage, since the underlying correlations can be easily modeled from measured data. Would that potentially reduce the fitting error, or better yet, simplify the operating procedure for lattices with larger numbers of sites?

Response: Reviewer is correct in highlighting that each eigenmode does not need to be controlled by an individual knob. In our system as well, this is indeed not the case. We want control knobs so that we can accurately modify the Hamiltonian parameters like onsite potentials to match the desired values. For this control part it is useful to have single knobs primarily affect one parameter at a time for ease of experiment. However, creating mapping functions that learn from data to control the eigen energies would indeed lead towards better control. In these cases, having low crosstalk helps in ensuring that the order and the number of correlation terms need to model does not get too large, and a proof of principle method based on this approach is what we present in the Fig. 3 of the paper where we use γ coefficients to model correlations in heater performance due to thermal crosstalk. More sophisticated data driven approaches, as highlighted in the above answer can then be adopted to reduce the error further in future works, but we want to highlight that having low crosstalk between knobs for control greatly helps in improving the accuracy of these predictive models as evident by our system where an illustrative model is able to achieve very accurate result.

REVIEWERS' COMMENTS

Reviewer #3 (Remarks to the Author):

As the authors have revised the manuscript, my questions have been answered and comments been addressed in detail. Particularly, from an implementation perspective, the new mode simulations demonstrating the behavior of thermo-optic heaters with and without the islands makes a much stronger case now. The potential reader can clearly see the details of the fabrication flow now, and the cross-sectional comparisons are also quite useful. The added discussion of applications for the demonstrated proof-of-concept device also conveys better context, and speaks to a wider audience in the updated version of the manuscript. I also appreciate the detailed answers regarding back-calculation of the hopping rate and the discussions regarding the quality factor of the resonators which have been added in the supplementary file.

In this current form, I recommend publication of the manuscript.

We are very grateful to the referee for reading our revised manuscript.

Reviewer #3 (Remarks to the Author):

Reviewer #3 (Remarks to the Author):

As the authors have revised the manuscript, my questions have been answered and comments been addressed in detail. Particularly, from an implementation perspective, the new mode simulations demonstrating the behavior of thermo-optic heaters with and without the islands makes a much stronger case now. The potential reader can clearly see the details of the fabrication flow now, and the cross-sectional comparisons are also quite useful. The added discussion of applications for the demonstrated proof-of-concept device also conveys better context and speaks to a wider audience in the updated version of the manuscript. I also appreciate the detailed answers regarding back-calculation of the hopping rate and the discussions regarding the quality factor of the resonators which have been added in the supplementary file.

In this current form, I recommend publication of the manuscript.

Response: We would like to thank the reviewer for a positive statement regarding our work and greatly appreciate the reviewer's help in the detailed review process and revision of the manuscript.